# A ubiquitous ice size bias in simulations of tropical deep convection

McKenna W. Stanford[1], Adam Varble[1], Ed Zipser[1], J. Walter Strapp[2], Delphine Leroy[3], Alfons Schwarzenboeck[3], Rodney Potts[4], and Alain Protat[4]

[1] Department of Atmospheric Sciences, University of Utah, Salt Lake City, Utah, 84112, USA
[2] Met Analytics, Inc., Aurora, Ontario, Canada
[3] Université Clermont Auvergne/CNRS, Laboratoire de Météorologie Physique, Clermont-Ferrand, France
[4] Australian Bureau of Meteorology, Research and Development Branch, Melbourne, Victoria, Australia

*Correspondence to*: McKenna W. Stanford (mckenna.stanford@utah.edu)

**Abstract.** The High Altitude Ice Crystals – High Ice Water Content (HAIC-HIWC) joint field campaign produced aircraft retrievals of total condensed water content (TWC), hydrometeor particle size distributions (PSDs), and vertical velocity ($w$) in high ice water content regions of mature and decaying tropical mesoscale convective systems (MCSs). The resulting dataset is used here to explore causes of the commonly documented high bias in radar reflectivity within cloud-resolving simulations of deep convection. This bias has been linked to overly strong simulated convective updrafts lofting excessive condensate mass but is also modulated by parameterizations of hydrometeor size distributions, single particle properties, species separation, and microphysical processes. Observations are compared with three Weather Research and Forecasting model simulations of an observed MCS using different microphysics parameterizations while controlling for $w$, TWC, and temperature. Two popular bulk microphysics schemes (Thompson and Morrison) and one bin microphysics scheme (Fast Spectral Bin Microphysics) are compared. For temperatures between -10 °C and -40 °C and TWC > 1 g m$^{-3}$, all microphysics schemes produce median mass diameters (MMDs) that are generally larger than observed, and the precipitating ice species that controls this size bias varies by scheme, temperature, and $w$. Despite a much greater number of samples, all simulations fail to reproduce observed high TWC conditions (> 2 g m$^{-3}$) between -20 °C and -40 °C in which only a small fraction of condensate mass is found in relatively large particle sizes greater than 1 mm in diameter. Although more mass is distributed to relatively large particle sizes relative to observed across all schemes when controlling for temperature, $w$, and TWC, differences with observations are significantly variable between the schemes tested. As a result, this bias is hypothesized to partly result from errors in parameterized hydrometeor PSD and single particle properties, but because it is present in all schemes, it may also partly result from errors in parameterized microphysical processes present in all schemes. Because of these ubiquitous ice size biases, the frequently used microphysical parameterizations evaluated in this study inherently produce a high bias in convective reflectivity for a wide range of temperatures, vertical velocities, and TWCs.

## 1 Introduction

Improving parameterizations is inherently challenging because they are often guided by quite crude observational constraints that fail to cover the large range of atmospheric conditions and cloud responses possible (Tao and Moncrieff, 2009; Khain et al., 2015). A common method used to evaluate the wide range of model physics schemes available is model intercomparison,

which involves rigorous comparison of cloud resolving model (CRM), limited area model (LAM), single column model (SCM), or general circulation model (GCM) output with high-quality in situ and remote-sensing observations, often from field experiments (e.g. Bechtold et al., 2000; Matsui et al., 2009; Wang et al., 2009a,b; Fridlind et al., 2010; Varble et al., 2011; Zhu et al., 2012; Varble et al., 2014a,b; and many others). Many resulting studies have focused on the characterization of cloud microphysical and dynamical properties in tropical deep convective systems (e.g. Stith et al., 2002, 2004; Heymsfield, 2003a,b;

Heymsfield et al., 2005, 2009, 2010; Lawson et al., 2015), which are climatically important because of their large contribution to global precipitation (Nesbitt et al. 2006). Ice microphysical properties are of particular interest because of anvil cirrus radiative effects (Hartmann et al., 1992; , Heymsfield and McFarquhar, 1996; McFarquhar and Heymsfield, 1996, 1997; Garrett et al., 2005) and the strong sensitivity of simulated deep convective systems to the parameterizations of species, habit, size distributions, and process rates (Chen and Cotton, 1988; McCumber et al., 1991; Gilmore et al., 2004; Milbrandt and Yau,

2005a; McFarquhar et al., 2006; Morrison and Grabowski, 2008).

This study utilizes data collected from the High Altitude Ice Crystals – High Ice Water Content (HAIC-HIWC) joint field campaign (Dezitter et al., 2013; Strapp et al., 2016). The first HAIC-HIWC phase was conducted in and around Darwin, Australia, from January to March in 2014 during the wet monsoon season. HAIC-HIWC was primarily designed to investigate meteorological processes responsible for commercial aircraft engine malfunction hypothesized to result from ingestion of high

ice water content ($> 2$ g m$^{-3}$) in regions of low radar reflectivity that did not raise alarm to pilots (Lawson et al., 1998, Mason et al. 2006). The campaign aircraft was thus equipped with instruments that measured particle cross-sectional areas from which particle size distributions (PSDs) were retrieved and total water content (TWC) from which condensate mass was retrieved in and around convective regions of tropical mesoscale convective systems (MCSs) where high ice water content (IWC) was expected to be encountered (Leroy et al., 2016a, hereafter referenced as L16).

The HAIC-HIWC datasets provide a wealth of in-cloud microphysical measurements that are ideal for evaluating simulations of tropical deep convective systems. One commonly documented model bias is the overestimation of radar reflectivity aloft in tropical deep convection, which has been shown to result from excessive graupel production in simulations (Blossey et al., 2007; Lang et al., 2007; Li et al., 2008, Matsui et al., 2009; Varble et al., 2011; Caine et al., 2013), but also from overly large snow sizes in some two-moment bulk microphysics schemes (Varble et al., 2011). Varble et al. (2014a)

explored the possible contribution of vertical velocity to this bias, and concluded that overly strong convective updrafts lofting large amounts of condensate mass in simulations were partially responsible for the reflectivity bias. However, they also found that the magnitude of this bias depends on the parameterization of microphysics. Franklin et al. (2016) compared a single-moment bulk microphysics scheme with HAIC-HIWC data and found that ice particle size aloft strongly impacts updraft

buoyancy and radar reflectivities in simulations of tropical deep convection, signaling the importance of interacting microphysics and dynamics. However, microphysical parameterization and convective dynamics contributions to the reflectivity bias have yet to be untangled, and individual contributions of hydrometeor type, size, and bulk mass to the bias have yet to be separated.

5       Lang et al. (2011, 2014) improved upon a single-moment bulk microphysics scheme by adjusting model physics for the purpose of reproducing observed reflectivity statistical distributions. However, their study was limited to using remote-sensing data, which provides limited information about vertical velocity and hydrometeor size, species, and bulk mass because of significant retrieval assumptions and uncertainties. In situ retrievals of vertical velocity, ice crystal size, and bulk mass during HAIC-HIWC provide a dataset needed to explore the various interacting contributors to model reflectivity biases. In

particular, mass-size distributions (MSDs) constructed from these measurements may be compared with simulated MSDs in microphysics schemes. L16 and Leroy et al. (2017) (hereafter referenced as L17) perform extensive analyses on median mass diameters (MMDs) retrieved from HAIC-HIWC data, a characteristic size metric that provides the diameter at which half of the bulk mass resides in smaller diameters and half resides in larger diameters. By comparing hydrometeor sizes and investigating the model parameters and processes that control size distributions, possible sources for biases may be identified

and further observational constraints on these parameters may be implemented.

        This study focuses on comparing simulated and observed hydrometeor sizes (e.g., MMDs) as functions of bulk mass and vertical velocity so that the role of microphysical processes and assumed particle properties in producing model convective precipitation biases can be isolated from the roles of total condensate and vertical velocity. Two bulk microphysics schemes and one explicit bin microphysics scheme are evaluated, providing insight into how the bias and its causes differ between

fundamentally different approaches to microphysics parameterization. The extent to which a reflectivity bias exists in bin microphysics schemes has not been extensively explored, although Ackerman et al. (2015) show that a CRM simulation using bin microphysics failed to reproduce observed low reflectivity values in high IWC regions, suggesting that this bias may exist across both bulk and bin schemes. Observations are described in Sect. 2, model setup and microphysics schemes in Sect. 3, intercomparison methodology and limitations in Sect. 4, results in Sect. 5, and conclusions in Sect. 6.

**2 Observations**

The majority of data collected during the first HAIC-HIWC field campaign was from instrumentation aboard the SAFIRE[1] Falcon 20 research aircraft. Strapp et al. (2016) describe the sampling strategy of the Falcon 20 during the HAIC-HIWC Darwin campaign, in which 17 out of 23 total flights targeted regions of large mature and decaying tropical MCSs with cold infrared (IR) brightness temperatures observed by satellite. Many flight legs penetrated convective updraft cores or regions

downstream of updraft cores at temperatures near -30 °C and -40 °C, with fewer flight legs performed around -50 °C and -10

---

[1] Service des Avions Francais Instrumentes pour la Recherche en Environnement

°C levels. Example flight tracks are shown in Fig. 1, overlaid on 10.8-µm IR brightness temperatures observed by the Japanese Multifunctional Transport Satellite 1R (MTSAT-1R) satellite, where the satellite image is a single time from when the system was sampled. MCSs sampled during Flight 6 on 23 January 2014 (Fig. 1a), Flights 12-13 from 2-3 February 2014 (Fig. 1b), Flight 16 on 7 February 2014 (Fig. 1c), and Flight 23 on 18 February 2014 (Fig. 1d) are shown because these are the events that were successfully simulated using the two bulk microphysics schemes. These events were chosen for simulation because of many observations at different temperature levels penetrating many convective updrafts with common occurrences of high IWC exceeding 2 g m$^{-3}$. Figure S1 shows the distribution of HAIC-HIWC samples by flight that are used in this study with simulated event dates labeled. The mean and standard deviation of temperature for each flight is also shown in Figure S1.

Vertical velocities ($w$) were calculated by SAFIRE using a method similar to that of Jorgensen and LeMone (1989), in which $w$ is defined as the difference between the vertical motions of the aircraft relative to the ground and relative to the air. The vertical motion with respect to air is calculated using the aircraft's true air speed along with attack, side-slip, pitch, and roll angles, the former two of which are measured using differential pressure measurements and the latter two of which use inertial navigation measurements. Uncertainty in $w$ calculations is estimated at ~ 1 m s$^{-1}$ (Jorgensen and LeMone, 1989).

Particle cross-section images used for derivations of PSDs were obtained by two optical array probes (OAPs): the 2D-Stereo probe (2D-S, Lawson et al., 2006) from SPEC Inc. and the Precipitation Imaging Probe (PIP, Baumgardner et al., 2011) from Droplet Measurement Technologies. The 2D-S measures particles with diameters ranging from 10-1280 µm at 10 µm resolution, and the PIP measures particles ranging from 100-6400 µm at 100 µm resolution. Linearly weighted composite PSDs for 5-second sampling windows were constructed using both OAPs, as described in L16. For these PSDs, truncated particle images were reconstructed using the method of Korolev and Sussman (2000). Based on spherical assumptions, truncated PIP images included in the PSDs derived by L16 reach diameters exceeding 12 mm. The OAPs were equipped with anti-shattering tips to avoid ice fragmentation (Korolev and Isaac, 2005; Heymsfield, 2007; Lawson, 2011) and an inter-arrival time algorithm was used to remove potentially shattered particles (Field et al., 2003, 2006; Korolev and Field, 2015).

Bulk TWC measurements were made with an isokinetic evaporator probe (IKP) first designed and developed by the National Research Council of Canada (NRC) and then re-engineered as the IKP2 by Science Engineering Associates (SEA) Inc. and NRC to operate within the constraints of the Falcon 20 aircraft (Davison et al., 2008; Strapp et al., 2016; Davison et al., 2016). The probe was designed specifically for the measurement of high altitude, high IWC conditions, with a target accuracy of 20%. The IKP2 uses a differential hygrometry method in calculating condensed water content (hereafter, TWC) that accounts for the subtraction of background water vapor. However, due to contamination of the background water vapor by ice crystals during the Darwin 2014 flight campaign, it was necessary to assume ice saturation inside cloud. There is therefore a fundamental TWC uncertainty due to the difference between the actual vapor concentrations in cloud versus the ice saturation estimate that may reach several tenths of a g m$^{-3}$ at -10 °C, dropping to about 0.1 g m$^{-3}$ at -40 °C. Work is continuing to better quantify this uncertainty, which is of course proportionately larger for low-IWC sections of cloud. Because of this uncertainty, observed samples with TWC $\leq$ 0.1 g m$^{-3}$ are excluded from this study. Further details about the

microphysical instrumentation used on the Falcon 20 and data processing from the OAPs and IKP2 may be found in L16, L17, and Strapp et al. (2016).

L16 use retrieved TWC with retrieved PSDs to constrain mass-size relationships ($m = \alpha D^\beta$) used to calculate MSDs over 5-second sampling intervals. The exponent $\beta$ is constrained through relation to the exponents of area-size and perimeter-size power law relationships derived from OAP images, which allows it to vary as a function of time as crystal habits change during flight. The parameter $\alpha$ is constrained by matching the integrated MSD to TWC measurements from the IKP2. The particle diameter size definition used for most results in this study is the 2D area equivalent diameter ($D_{eq}$), defined as the diameter of a circle with the same area as particle images from the OAPs. However, some statistics are shown using the $D_{max}$ definition, defined here as the maximum dimension through the center of the 2D crystal image.

Mass-size distributions, defined as the product of the PSD, given by $N(D)$, and the mass-size power law relationship, are given by Eq. (1):

$$M(D) = \alpha D^\beta N(D) \tag{1}$$

where $M(D)$ has units of kg m$^{-4}$. Percentiles of the MSD are used to describe hydrometeor size, and are calculated by numerically integrating the MSD from 0 to the mass diameter (MD) where the integrated mass equals the desired percentage of total mass. In particular, the MMD is used for comparison of retrieved and simulated hydrometeor sizes, where the MMD is calculated using Eq. (2):

$$\int_0^{MMD} \alpha D^\beta N(D) dD = \frac{1}{2} \int_0^\infty \alpha D^\beta N(D) dD = \frac{1}{2} TWC \tag{2}$$

Importantly, we note that large uncertainties exist in retrievals of total number concentration for diameters smaller than 150 µm based on the derivation technique employed (not shown). However, TWC and MSD measurements that are based on higher moments of the PSD are more certain since the IKP2 measures TWC plus water vapor, where the subtraction of water vapor is the primary source of uncertainty, and MSDs are not strongly impacted by particles smaller than 150 µm (L17). Moreover, the MSD dataset derived by L16 permits comparison of simulated and observed hydrometeor properties in the context of TWC and $w$ that are more accurate and detailed than remote-sensing retrievals. However, because an objective of this study is to investigate well-known reflectivity biases, data from a C-band scanning dual-polarimetric radar (C-POL) (Keenan et al., 1998) located near Darwin are utilized for Flight 23 on 18 February 2014, one of the only events that occurred within range of the radar. The C-POL dataset for 18 February permits establishing reflectivity biases in the current study and provides motivation for focusing on this particular simulated event for comparison with observed microphysical quantities.

### 3 Model

### 3.1 Model Setup

The Advanced Research Weather Research and Forecasting (WRF-ARW) V3.6.1 model (Skamarock et al., 2008) is used to simulate the MCS events shown in Fig. 1 using two different bulk microphysics schemes, while the 18 February event is also

simulated using a bin microphysics scheme. The Bureau of Meteorology's (BoM) Australian Community Climate and Earth-System Simulator Regional model (ACCESS-R) (Puri et al., 2013) analyses with 12-km horizontal grid spacing are used for initial and boundary conditions. Physics parameterizations common to all simulations performed include the Mellor-Yamada-Janjic (MYJ) planetary boundary layer (PBL) scheme (Janjic, 1994), the Rapid Radiation Transfer Model (RRTM) longwave

radiation scheme (Mlawer et al,. 1997), the Dudhia (1989) shortwave radiation scheme, the Kain-Fritsch cumulus scheme (Kain, 2004), and the Noah Land Surface Model (Chen and Dudhia, 2001). All of the simulated MCS events use 9:3:1-km two-way nesting with 92 vertical levels and the innermost 1000-m grid spacing domain is used for analysis. The nested domains and C-POL coverage for the 18 February simulations are shown in Fig. 2.

## 3.2 Microphysics

The microphysics schemes employed in this study include the Thompson (Thompson et al., 2008) and Morrison (Morrison et al., 2009) bulk microphysics schemes and the Hebrew University Fast Spectral Bin Microphysics (FSBM) scheme (Lynn et al., 2005). The Thompson and Morrison bulk schemes predict integral moments of the PSD for five hydrometeor species, including cloud ice, cloud water, rain, snow, and graupel. PSDs in both bulk schemes are generally represented by a gamma function of the following form:

$$N(D) = N_0 D^\mu e^{-\lambda D} \tag{3}$$

where $N_0$ is the intercept parameter, $D$ is the particle diameter, $\mu$ is the shape parameter, and $\lambda$ is the slope parameter. $N_0$ essentially controls the number of small particles, $\mu$ controls the dispersion of the PSD, and $\lambda$ controls the slope of the PSD. Snow, graupel, rain, and cloud ice are double-moment species in the Morrison scheme with both mass mixing ratio ($q$) and number concentration ($N$) predicted, whereas cloud water is a single-moment species (prognostic $q$ only and constant $N$). The Thompson scheme uses double-moment rain and cloud ice with single-moment snow, graupel, and cloud water. An exception

to the generalized gamma distribution given by Eq. (3) is the Thompson snow PSD parameterization, which uses a bimodal gamma distribution given by Field et al. (2005) that varies with temperature. The Morrison scheme assumes a constant bulk density for all ice species given by Reisner et al. (1998), while Thompson only assumes constant graupel and cloud ice density. For snow, the mass-size power law relationship presented in Cox (1988) is used that allows the bulk density of snow to vary with particle size. Varble et al. (2014a) showed that this relationship (where $m \propto D^2$) reproduces observed reflectivity better

than schemes assuming $m \propto D^3$ for snow, while others have shown that it is supported by surface disdrometer (Mitchell et al., 1990) and aircraft (Westbrook et al., 2004) observations of snow particles. While the Thompson scheme uses a constant bulk density typical of medium density graupel, it varies $N_0$ inversely as a function of the predicted $q$ and shifts the fall-speed relationship from graupel toward hail as particle size increases. For the versions of the Morrison and Thompson schemes used in this study, $\mu = 0$ for rain, graupel, and cloud ice, making the PSD exponential. For snow, $\mu = 0$ for Morrison, but is nonzero

in the Thompson scheme. Cloud ice $\mu$ is also variable in both schemes. Lastly, this study uses a constant cloud droplet number concentration of 100 cm⁻³ in both bulk schemes that is typical of the clean tropical maritime air masses commonly observed in

the vicinity of Darwin during the active monsoon. A summary of the PSD, mass-size, and terminal velocity-size relationship parameters used in the bulk schemes may be found in Tables 1 and 2.

The FSBM scheme uses 33 mass-doubling bins to represent PSDs and process rates are computed separately for each bin. FSBM species include cloud condensation nuclei (CCN), liquid water (both raindrops and cloud droplets), graupel, and ice crystals/aggregates (hereafter referred to as snow). Graupel has a bulk density of 400 kg m$^{-3}$ and liquid has a bulk density of 1000 kg m$^{-3}$. The density of vapor-grown ice in FSBM varies from 900 kg m$^{-3}$ at small sizes to 35 kg m$^{-3}$ at large sizes. It also explicitly represents cloud droplet nucleation whereas the bulk schemes do not, and it is able to maintain supersaturations over liquid that the bulk schemes cannot. Initial CCN concentrations in the FSBM scheme are also set to resemble the maritime environment in Darwin using FSBM's default maritime airmass aerosol concentration, in which concentrations near the surface are ~ 100 cm$^{-3}$ and decrease exponentially with height to ~ 50 cm$^{-3}$ at 4-km altitude and < 10 cm$^{-3}$ at 9-km altitude. A summary of FSBM mass-size and terminal velocity-size relationships are given in Table 3.

## 4 Intercomparison Methodology

Simulating every HAIC-HIWC event with multiple model setups is not computationally feasible, and therefore, four MCS events shown in Fig. 1 are simulated using the Morrison and Thompson microphysics schemes. The primary differences between simulated events are the mesoscale precipitation structure and peak convective intensities, while simulated hydrometeor properties vary little between events when controlling for $w$ and TWC (see Section 5.2.2 and respective discussion of Figures S2-S4). Much larger differences exist between simulations with different microphysics schemes when simulating the same case, suggesting that a single simulated event is adequate for robustly examining differences between various microphysics schemes and observations. Moreover, Figure S1 shows that using observations from the entire Darwin field campaign is necessary in order to stratify MMD and TWC by temperature with sufficient sample sizes. The 18 February event (Flight 23) contains the most observations in high TWC conditions near -10 to -15 °C and is the only flight within observing range of C-POL, two primary reasons that simulations of it are used for comparison with observations. The 450-km by 540-km inner domain (1000-m grid spacing) for this case is shown in Fig. 2. The simulation was run for 30 hours from 00Z on the 18[th] to 06Z on the 19[th]. Observations from all HAIC-HIWC flights are compared with results from the innermost domain for a 6-hour period between 18Z on the 18[th] and 00Z on the 19[th] during the mature and decaying stages of the MCS.

Four primary variables are analyzed: temperature (T), TWC, $w$, and percentiles of the MSD (i.e., 10% MD, MMD, and 90% MD). Observed MDs and bulk mass are not separated by species. However, L16 state that only trace amounts of liquid water content (LWC) were detected for a few flights at T > -20 °C, and thus, TWC is a reasonable proxy for IWC in the vast majority of observed situations. Relatively small amounts of LWC measured in the mature and decaying MCSs sampled during HAIC-HIWC differ from measurements in isolated, growing cumulus cells such as those sampled during the Ice in Clouds Experiment – Tropical field campaign, which show a considerable amount of supercooled liquid at temperatures down to -15 °C (Heymsfield and Willis, 2014; Lawson et al., 2015; Yang et al., 2016). Simulated bulk mass and MDs are calculated

for both individual and combined species, where the combined-hydrometeor MSD, given by $M(D)_{tot}$, is the composite MSD consisting of all hydrometeor species in the scheme, as given by Eq. (4):

$$M(D)_{tot} = \sum_{i=1}^{n} \alpha_i D^{\beta_i} N_i(D) \qquad (4)$$

where $n$ is the number of species in the microphysics scheme. For the evaluation of mass partitioning between species in bulk schemes, liquid MSDs combine cloud water and rain while snow MSDs consist of all vapor grown ice: cloud ice and snow.

5        Comparison of simulation output and measurements are confined to flight segments and grid points containing TWC $> 0.1$ g m$^{-3}$ in order to avoid large observational uncertainty associated with smaller TWC values. Comparisons are also limited to -60 °C $\leq$ T $\leq$ 0 °C. With these constraints, grid point sample sizes for individual simulated events are greater than $10^6$, approximately 2 orders of magnitude greater than the observational sample size. While TWC and MDs are analyzed for both positive and negative vertical velocities, emphasis is placed on updrafts since the production of high IWC requires

condensation of a substantial amount of water vapor, whereas adiabatic warming and drying in downdrafts typically counteracts this process and leads to evaporation. However, since regions of high IWC certainly exist outside of updrafts after they are detrained, downdrafts are included to determine possible ice size biases in both types of convective motions. Along with the bulk mass constraint, "updrafts" are defined as points with $w > 1$ m s$^{-1}$, "downdrafts" as points with $w < 1$ m s$^{-1}$, and points with $w$ between -1 and 1 m s$^{-1}$ are considered relatively quiescent regions.

A possible comparison bias may result from different grid spacing in simulations and observations. Simulated events are analyzed within the 1000-m horizontal grid spacing domain, whereas observed PSDs are retrieved using 5-second sampling windows, which corresponds to a grid spacing of ~ 750 m assuming a typical aircraft speed of 150 m s$^{-1}$. However, similar results between a 333-m grid spacing domain simulation and the 1000-m domain for the 18 February event (not shown) suggest that this grid spacing difference does not significantly contribute to large differences between simulations and observations.

The most significant source of comparison bias is the subjective observational sampling. Regions with lightning or "red" on the pilot's X-band radar display (reflectivity exceeding 40 dBZ) were avoided during flights, and these regions likely contain the most intense convective cells with the most graupel and liquid water (e.g., Zipser and Lutz, 1994). This sampling cannot be replicated in simulations because of the previously mentioned biases in simulated reflectivity and the lack of simulated lightning. Reflectivities exceeding 40 dBZ were infrequent at flight level for most flights, with the exception of legs

at temperatures near -10 °C, however lightning was common in the most intense convective cells associated with many of the large MCSs. Possible effects of this bias on interpretation of results are discussed further in subsequent sections.

## 5 Results

### 5.1 Radar reflectivity

Simulated Rayleigh radar reflectivity ($Z_e$) for bulk schemes is calculated by integrating over the sixth moment of the melted equivalent diameter ($D_{eq,melt}$) size distribution for each individual hydrometeor species and summing $Z_e$ for all species. For ice particles, $D_{eq,melt}$ is given by Eq. (5):

$$D_{eq,melt} = \left[\frac{6\alpha}{\pi\rho_w}\right]^{\frac{1}{3}} D^{\frac{\beta}{3}} \tag{5}$$

where $\rho_w$ is the bulk density of water. Reflectivity for bulk scheme ice hydrometeors is then given by Eq. (6):

$$Z_e = 0.224 \times 10^{18} \int_0^\infty D_{eq,melt}^6 \, N\left(D\{D_{eq,melt}\}\right) \left|\frac{\partial D}{\partial D_{eq,melt}}\right| dD \tag{6}$$

where $10^{18}$ is a conversion factor from m$^6$ to mm$^6$ and 0.224 is a factor accounting for the different dielectric constants of ice and liquid, following Smith (1984). For liquid water species in bulk schemes, $D_{eq,melt} = D$ and Eq. (6) reduces to Eq. (7):

$$Z_e = 10^{18} \int_0^\infty D^6 N(D) \, dD \tag{7}$$

Recall that the FSBM scheme does not assume a continuous PSD but rather computes number concentrations for discrete particle size (and mass) bins. Therefore, FSBM reflectivity is calculated for each discrete bin using the integrands of Eq. (6) and Eq. (7) and then summed to give a total reflectivity. Another important difference between reflectivity calculations in FSBM and bulk schemes is the use of a parameterization in bulk schemes to account for partially melted ice coated with water (Blahak, 2007) which is not used in FSBM, however this does not impact our analyses since they focus on sub-freezing temperatures. The presented (Rayleigh) approximation for simulated reflectivity works well for comparison with C-POL's wavelength (5.5-cm) in convective systems without a significant amount of large hail as is expected during Darwin's active monsoon period. C-POL reflectivity is interpolated to a Cartesian grid with a 1-km horizontal grid spacing and 500-m vertical grid spacing. Simulated reflectivity used for comparison maintains the native 1-km horizontal grid spacing but is interpolated vertically to C-POL's constant altitude levels for comparisons.

Observed and simulated Rayleigh radar reflectivity horizontal cross-sections are shown for the 18 February 2014 MCS at 18Z in Fig. 3-4 for 2.5-km (~ 13 °C) and 7-km (~ -10 °C) altitudes, respectively. Observed 2.5-km altitude reflectivities are typically 40-45 dBZ in convective cores with an expansive stratiform region containing reflectivities between 25-35 dBZ. Both bulk schemes produce much more widespread regions of reflectivity exceeding 40 dBZ with convective core values reaching 55 dBZ and more convective organization than observed. The FSBM scheme produces reflectivity values more closely to those observed, although the system is somewhat less organized than observed and produces lesser maximum reflectivities than C-POL at this altitude. At 7-km (Fig. 4), observed reflectivities mostly remain below 25 dBZ, with values reaching 30-35 dBZ for only the most intense reflectivity cores. Conversely, all simulated reflectivities at 7-km exceed 25 dBZ in much more expansive regions compared to C-POL. Thompson produces a cellular structure with cores exceeding 55 dBZ,

and Morrison and FSBM produce widespread regions with reflectivities > 40 dBZ. Fig. 3-4 clearly show differences in both the horizontal and vertical precipitation structures of the simulated and observed MCSs, and changing the microphysics parameterization also produces substantially different results.

More is revealed through examination of vertical reflectivity profiles. Figure 5 shows 99[th] percentile profiles of reflectivity $\geq$ 5 dBZ for the 18 February MCS during two time periods: 12Z-18Z on the 18[th] (Fig. 5a) and 18Z on the 18[th] to 00Z on the 19[th] (Fig. 5c). Sample sizes normalized by domain area are shown in Fig. 5b and Fig. 5d for the 12Z-18Z and 18Z-00Z time periods, respectively. Two time periods are shown because the most intense observed convection occurred before 18Z, but the flight took place after this time when the most intense convection had moved out of the C-POL domain. During the 12Z-18Z time period, all simulated 99[th] percentile reflectivities exceed C-POL 99[th] percentile reflectivities by 10-15 dBZ above the melting level throughout the majority of the free troposphere. Differences between C-POL and simulations are larger in the 18Z-00Z time period where the simulated 99[th] percentile reflectivities exceed observed 99[th] percentile reflectivities by up to 20 dBZ above the melting level. These large discrepancies in both time periods suggest a significant reflectivity bias at sub-freezing temperatures. The Thompson scheme best reproduces the observed reflectivity vertical gradient, but absolute values of the reflectivity profiles aloft are closer between the simulations than between a single simulation and observed values. Additionally, there is no clear advantage in using bin versus bulk schemes for reproducing observed reflectivity profiles.

### 5.2 MMD-T-*w*-TWC relationships

Isolating the role of potential simulated hydrometeor size biases in producing reflectivity biases requires controlling for *w* and TWC, which can also be biased and impact reflectivity. This is accomplished by examining relationships between MMD, *w*, T, and TWC.

### 5.2.1 Observations

Figure 6a shows average TWC (color-fill) as a function of T (ordinate) and *w* (abscissa) bins, while Fig. 6b and 6c show average MMD (color-fill) as a function of *w*-T and TWC-T bins, respectively. Note that the MMD color-fill scale is nonlinear. Observed mean TWCs in Fig. 6a range from 1 to 3 g m$^{-3}$ and generally increase with increasing *w* for updrafts and increase with decreasing *w* for downdrafts. There is no clear relationship between TWC and T for T > -25 °C, although this is likely a result of biased observational sampling at warm temperatures. For T < -25 °C, TWC tends to increase for increasing T. Figure 6b shows a clear T-dependency for mean MMDs, which generally decrease with decreasing T. Observed mean MMDs range from 300-400 µm at T < -30 °C and approach ~1 mm at -10 °C. Most observed MMDs at T < -20 °C decrease slightly with increasing *w* for updrafts, but due to small sample sizes, no conclusions can be drawn for warmer T. MMD also appears to decrease slightly with downdraft velocity for a given T and are in fact higher for *w* between -1 and 1 m s$^{-1}$ compared to downdrafts, although downdraft samples are primarily limited to *w* > -7 m s$^{-1}$. Figure 6c also shows that for T < -30 °C, MMDs generally decrease with increasing TWC, agreeing with L17, who show that observed MMDs in high IWC regions decrease with decreasing T and increasing TWC. However, this MMD-T-TWC relationship is sensitive to the type of MCS sampled.

L17 discuss MMDs increasing with increasing TWC for Flights 12 and 13 (see Fig. 1) that sampled a long-lived, strongly cyclonic tropical low with persistent very deep convection at the center of the circulation. This inverse relationship is visible in Fig. 6c for T between -32 °C to -36 °C and between -24 °C and -28 °C. Interrogation of OAP images by L17 suggest that the dominant growth process for the larger MMDs during this event is vapor deposition, consistent with modest vertical wind speeds and little to no lightning during the event.

### 5.2.2 Differences between observations and simulations

Figure 7 shows relative differences between observed and simulated (simulated minus observed) mean TWC as a function of $w$-T bins (7a-c), as well as relative differences between observed and simulated mean MMD as a function of $w$-T (7d-f) and TWC-T (7g-i) bins. Differences between Thompson and observations are shown in the left column, between Morrison and observations in the middle column, and between FSBM and observations in the right column. Figure 7a-c shows that all simulations generally produce lesser than observed TWC for T < -24 °C and $w$ between -5 and 7 m s$^{-1}$, but generally larger TWC for warmer T, $w > 7$ m s$^{-1}$, and a few downdraft bins. The Thompson scheme reproduces observed mean TWCs with the greatest accuracy, generally remaining within 50% of those observed. Besides a few downdraft bins, the Morrison and FSBM schemes produce lesser TWC than observed by up to 50-100% at T < -32 °C and $w < 5$ m s$^{-1}$. However, for T > -30 °C or $w >$ 8 m s$^{-1}$, Morrison and FSBM produce slightly greater TWC than Thompson and observations. Recall that observed TWC uncertainty is ~0.1 g m$^{-3}$ at -40 °C and ~0.3 g m$^{-3}$ at -10 °C, however any potential bias over the entire range of samples is likely smaller and analyses of TWC differences between observations and simulations in this study focus on larger TWC values greater than 1 g m$^{-3}$.

Figure 7d-f shows relative differences between simulated and observed mean MMDs as a function of $w$ and T, where the simulated MMDs include all hydrometeor species together to mimic observations (Eq. 4). For T < -30 °C and $w < 8$ m s$^{-1}$, the Thompson scheme generally produces smaller than observed mean MMDs, whereas the Morrison and FSBM schemes produce larger than observed mean MMDs for the same $w$-T bins. Morrison produces the largest mean MMDs, commonly exceeding 100% larger than observed. For T > -30 °C, $w > 7$ m s$^{-1}$, and $w < -1$ m s$^{-1}$, all schemes produce significantly larger mean MMDs than observed. All schemes exhibit a minimum in relative differences with observations for $w$ between -1 m s$^{-1}$ and 1 m s$^{-1}$ and increasing relative differences with increasing absolute value of $w$.

Figure 7g-i also shows relative differences between observed and simulated MMDs, but the abscissa is TWC rather than $w$. The Thompson scheme produces smaller than observed MMDs for T < -30 °C and TWC < 2 g m$^{-3}$. For T > -24 °C and TWC > 0.5 g m$^{-3}$, or for T between -24 °C and -40 °C and TWC > 2 g m$^{-3}$, Thompson produces larger than observed mean MMDs. The FSBM scheme produces larger than observed mean MMDs across almost every TWC-T bin, but generally remains within 50% of observations for TWC < 1 g m$^{-3}$, while differences are exacerbated for larger TWC > 1 g m$^{-3}$. Morrison again produces the largest mean MMDs, exceeding 100% larger than observed across the majority of TWC-T bins.

Figures S2-S4 in the supplemental material show differences between observations and the three additional HIWC events (23 January, 2-3 February, and 7 February, see Fig. 1) simulated using the bulk schemes. Results from these additional

simulations are very similar to those shown in Figure 7 for the 18 February event. This provides justification for using a single simulated event for comparison with the observations from all HIWC flights. Moreover, Figures S2-S4 show that there is little variability in MMD-$w$-T-TWC relationships as a function of simulated event for a given bulk microphysics scheme.

Figure 7 suggests that high biases in simulated radar reflectivity are partly a result of too much condensate mass residing in particle diameters that are greater than observed MMDs of 0.4 to 1 mm rather than the lofting of excessive condensate mass and large particles by exaggerated vertical velocities alone. Proving that simulations have a hydrometeor size bias is difficult because of biased observational sampling that avoided the highest reflectivity convective cores. However, including sampling of these cores would still fail to bring observed and simulated MDs together for a given $w$ or TWC since observed reflectivities aloft are significantly less than simulated, as was shown in Fig. 5. Additionally, analyzing minimum 90% MDs as a function of $w$ or TWC can definitively establish a model hydrometeor size bias. The minimum 90% MD is defined as the minimum value among the distribution of 90% MDs within a given TWC-T or $w$-T bin. The minimum 90% MD is analyzed because it should not be impacted by the observational sampling bias (avoidance of lightning and reflectivities exceeding 40 dBZ) since it is associated with a lack of large, dense particles, whereas the observed MMDs shown in Figure 7 may be partly impacted by this observational sampling bias. If simulations fail to reproduce at least the minimum of the observed 90% MD distribution for a given bin, then there is greater confidence that a model bias exists. Figure 8 shows joint histograms of simulated minus observed minimum 90% MDs as a function of TWC-T bins for bins where simulated values are greater than observed values. Simulations have a factor of $10^2$ more samples than observations, and therefore, bins with larger simulated than observed values almost certainly have 90% MDs that are biased high, implying again that too much mass is distributed to large particle sizes in simulations. Figure 8a shows that minimum 90% MDs are larger in the Thompson scheme for TWC > 1 g m$^{-3}$ and T between -24 °C and -40 °C or for TWC > 2 g m$^{-3}$ and T between -8 °C and -20 °C. The Morrison scheme (Fig. 8b) again produces the greatest differences with most bins for T between -20 °C and -50 °C and TWC > 0.5 g m$^{-3}$ exhibiting relative differences of 100% or greater. For T between -8 °C and -20 °C and TWC > 1 g m$^{-3}$, Morrison produces relative differences between 50% and 100 %. FSBM (Fig. 8c) produces larger than observed minimum 90% MDs for T between -16 °C and -50 °C and most TWC bins, although the relative difference magnitudes are smaller relative to Thompson and Morrison.

## 5.3 MD-$w$-TWC relationships in specific temperature ranges

Figure 9a-c shows average 10% MDs, MMDs, and 90% MDs, respectively, as functions of $w$ for T between -32 °C and -40 °C. The Morrison and FSBM schemes produce larger than observed 10%, 50%, and 90% MDs for all $w$, with differences as large as 3 mm for the 90% MDs. The Thompson scheme is the only scheme that produces smaller than observed 10% MDs, whereas it captures observed MMDs remarkably well for $w$ between 4 m s$^{-1}$ and 10 m s$^{-1}$, although it diverges from observations outside of this range. Although observed MMDs at larger $w$ values may be biased low due to observational sampling that avoids the most intense convective cores, high biased Thompson reflectivities in this T range (see Fig. 5) suggest more representative observational sampling would not erase the MMD difference. This is even more evident for Thompson 90%

MDs in Fig. 9c, which reach values as high as 8 mm for 20 m s$^{-1}$ updrafts. Similar to Thompson, Morrison increases MMDs and 90% MDs with $w$ for updrafts, but limits 90% MDs to less than 4 mm. FSBM produces an entirely different 90% MD-$w$ relationship than the bulk schemes, where 90% MDs increase with increasing $w$ between 1 m s$^{-1}$ and 6 m s$^{-1}$, but decrease with increasing $w$ above this threshold. Similar relationships exist between MDs and TWC as for MDs and $w$ in Fig. 9, a result of TWC generally increasing with increasing $w$ in this T range (see Fig. 6a and 11a-c in Sect. 5.4).

Figure 10 shows the same information as Fig. 9, but for a T range between -8 °C and -16 °C. Observations in Fig. 10 are plotted as individual data points (black diamonds) because of limited sampling of intense updrafts/downdrafts for T > -20 °C during the Darwin campaign. Limited measurements prevent conclusions about observed MD-$w$ relationships in this T range, however some inferences can be drawn. Note that much larger hydrometeor sizes are present than at colder temperatures, and thus the ordinate in Fig. 10 is larger than in Fig. 9 for MMDs and 90% MDs. Observed 10% MDs generally decrease with increasing $w$ for updrafts, a relationship that all schemes capture to varying degrees. The Thompson scheme reproduces the observed 10% MD-$w$ relationship best for $w$ between 0 m s$^{-1}$ and 4 m s$^{-1}$, but produces smaller than observed 10% MDs for higher $w$, related to excessive cloud droplet production. Morrison also produces smaller than observed 10% MDs for $w$ > 4 m s$^{-1}$, but larger 10% MDs for smaller $w$. FSBM has far fewer cloud droplets (see Section 5.4) and produces larger than observed 10% MDs in updrafts and downdrafts. Thompson and Morrison MMDs increase with increasing $w$ and are larger than observed outside of quiescent regions, although Thompson produces much greater MMDs compared to Morrison for $w$ >10 m s$^{-1}$. Thompson 90% MDs increase with increasing $w$ for updrafts while FSBM 90% MDs decrease with $w$ for $w$ > 2 m s$^{-1}$. Morrison 90% MDs increase with increasing $w$ between 0 m s$^{-1}$ and 6 m s$^{-1}$, but plateau at 6 mm for $w$ > 6 m s$^{-1}$. While observed sample sizes in this temperature range are too small to confidently establish a 90% MD-$w$ relationship, most observed 90% MDs are smaller than simulated and 90% MDs appear to increase with increasing $w$ in updrafts. Observed MDs as a function of TWC do not show a significant relationship in this temperature range, and thus MD-TWC relationships are not discussed here.

**5.4 Model hydrometeor species partitioning**

Recall that simulated MDs are computed from a composite MSD that incorporates all hydrometeor species in each microphysics scheme (Eq. 4). Therefore, differences between simulations and observations shown in Fig. 7-10 may be partially explained by how mass is partitioned among species with different properties and how particle size is distributed by mass (MSD) for an individual hydrometeor species. In bulk schemes, these distributions are strongly modulated by the PSD parameters ($N_0$, $\mu$, and $\lambda$) and the number of prognostic PSD moments (Varble et al., 2011). Although fewer assumptions are made regarding PSDs in the FSBM scheme, separate hydrometeor species with parameterized particle properties and microphysical processes still contribute to potential model biases. This section examines how MSD parameters (Eq. 1), assumed particle properties, and microphysical processes within each scheme may affect this partitioning of mass and particle size.

Figure 11 shows $w$-T joint histograms for the 18 February 2014 simulated MCS. The color-fill in panels (a)-(c) is average TWC for Thompson, Morrison, and FSBM, respectively, and the color-fill in panels (d)-(f) is the average combined-

hydrometeor MMD. Figures 11a-c show that all schemes produce increasing TWC with increasing T and increasing downdraft or updraft velocity. The Morrison scheme produces the largest TWC with values of 4 g m$^{-3}$ or greater for $w > 15$ m s$^{-1}$ and T > -30 °C. Each scheme produces substantially different distributions of MMD as a function of T and $w$ (Fig. 11d-f), which is revealing when examined in conjunction with bulk mass. Both bulk schemes increase MMD with increasing T and increasing downdraft or updraft velocity. FSBM increases MMD with increasing T and decreasing updraft strength for many bins, a relationship also exhibited by observations (Fig. 6b). Thompson produces the largest combined-hydrometeor MMDs, with maximum values approaching 1 cm for T > -10 °C and $w > 20$ m s$^{-1}$. Despite this, it shows a sharp T-dependency with mean MMDs smaller than 400 µm at T < -40 °C across all $w$ values. Morrison average MMDs range from 1-3 mm for the majority of $w$-T bins, producing the largest sizes of any scheme for T < -30 °C, a feature also shown in Fig. 9b. FSBM clearly produces the smallest MMDs on average, reaching a maximum of ~ 2 mm for weak $w$ values and warm T.

Figure 12 shows joint histograms organized similarly to Fig. 11, but the color-fill in panels (a)-(c) is average snow (combined with cloud ice) water content (SWC), and the color-fill in panels (d)-(f) is average snow MMDs. The FSBM scheme produces the largest SWC with values of 1.5-2 g m$^{-3}$ for T between -20 °C and -40 °C and $w > 10$ m s$^{-1}$. Morrison produces the lowest SWC, remaining below 1 g m$^{-3}$ for all $w$-T bins, whereas Thompson produces values of 1.5-1.75 g m$^{-3}$ for temperatures between -20 °C and -40 °C. The T-dependency in the Thompson snow MMDs is clear with a maximum of 1-3 mm for T > -20 °C and $w < 10$ m s$^{-1}$, decreasing to smaller than 0.5 mm at most T < -30 °C across all $w$. This pattern results from the T-dependent Thompson snow PSD parameterization coupled with the *m-D* relationship in Table 1 that forces a small, dense particle mode at colder T. Thompson's large SWC for T < -40 °C in Fig. 12a largely controls the combined-hydrometeor MMD at these temperatures (Fig. 13d) and is the cause for smaller than observed MMDs shown in Fig. 7 and smaller than observed 10% MDs for cold temperatures in Fig. 9. The FSBM snow MMD relationship with T and $w$ is somewhat similar to Thompson since it also uses a variable snow density that accounts for high-density small crystals and low-density large aggregates. However, FSBM snow particles are not diagnosed to be small at cold T like they are in Thompson, and it produces MMDs up to 0.5 cm just above the melting level. Morrison produces the largest mean snow MMDs of all schemes across most $w$-T bins, with sizes approaching 1 cm for T > -10 °C and $w > 5$ m s$^{-1}$.

Figure 13 is similar to Fig. 12, but shows graupel water content (GWC) and graupel MMDs. All schemes increase graupel MMDs and GWC with increasing T and increasing downdraft or updraft velocity. Thompson and FSBM GWC are distributed similarly, although FSBM generally produces more GWC than Thompson for a given $w$-T bin. Morrison produces the largest GWCs, which account for most of the Morrison TWCs shown in Fig. 11b, strongly modulating Morrison combined-hydrometeor MMDs in Fig. 11e. Thompson produces the largest graupel sizes of all the schemes, with many average graupel MMDs > 1 cm for high $w$ and warm T. These large graupel sizes are a result of the diagnostic inverse relationship between graupel mass and $N_0$, forcing graupel to larger sizes as GWC increases (see Table 1). The Thompson scheme also shifts the fall-speed relationship to be more representative of hail at larger graupel sizes, resulting in faster sedimentation and limited GWC at cold T or small $w$ (Fig. 13a). However, this also creates very large graupel particles in intense updrafts in which graupel continues to be carried upward, which biases the combined-hydrometeor MMD, as was shown in Fig. 7, 9, and 10.

Morrison produces smaller graupel MMDs than snow MMDs (cf., Fig. 12e and 13e), where graupel sizes generally remain around 1-3 mm for most $w$-T bins. However, smaller graupel MMDs in conjunction with a slower terminal fall-speed results in less graupel sedimentation in the Morrison scheme and thus higher GWC than in the Thompson scheme. The Morrison combined-hydrometeor MMD (Fig. 11e) is largely controlled by graupel because of the large GWCs, despite snow being the

largest precipitating ice species because of low SWCs. FSBM produces the smallest mean graupel MMDs of all the schemes, generally remaining below 1 mm for most $w$-T bins and reaching sizes of ~2 mm just above the melting level. FSBM graupel MMDs are even smaller than FSBM snow MMDs (cf., Fig. 12f and 13f) and smaller than bulk scheme graupel MMDs. The unique FSBM feature of decreasing MMD with positive $w$ is caused by the graupel MMDs being smaller than snow MMDs coupled with decreasing SWC and increasing GWC with increasing updraft strength.

Average LWC (combining cloud water and rain) and liquid MMD joint histograms are shown in Fig. 14. The bulk microphysics schemes distribute LWC similarly to one another as a function of T and $w$, although Thompson produces slightly more LWC than Morrison, especially just above the melting level. The FSBM scheme produces the largest LWCs for T between 0 °C and -8 °C, but much less supercooled LWC compared to bulk schemes for T < -8 °C. Thompson and Morrison liquid MMDs are relatively similar to one another, with cloud droplets dominating at T < -4 °C and raindrops dominating for

T > -4 °C. Recall that for both bulk schemes, rain is a double-moment species and cloud water is a single-moment species. The FSBM scheme produces far different liquid MMDs as a function of $w$ and T compared to the bulk schemes. For T > -4 °C, FSBM produces the smallest mean liquid MMDs and largest LWCs. For T < -4 °C, FSBM produces liquid MMDs more typical of small drizzle drops than cloud droplets. These unique features in the FSBM scheme may be the result of the explicit activation of CCN and the maintenance of liquid supersaturation in FSBM that are absent in the bulk schemes. Aerosol

consumption below the melting level in FSBM may cause faster and more efficient collision-coalescence than in the bulk schemes, allowing large raindrops to sediment more easily before reaching sub-freezing T. The smaller raindrop MMDs in FSBM may also be partly responsible for the smaller FSBM mean graupel MMDs compared to the bulk schemes (see Fig. 13f). Figures 14c and 14f show that raindrop-sized MMDs control the large LWCs between 0 °C and -4 °C in updrafts, and IWC is negligible in updrafts at these temperatures. By -8 °C, nearly all liquid in updrafts is gone while GWC has increased

dramatically, much more so than SWC (cf. Fig. 13f and Fig. 14f). Furthermore, GWC decreases with decreasing T below -8 °C, even though FSBM graupel terminal fall speeds (< 4 m s$^{-1}$) are weaker than the updraft speeds so that the graupel particles are carried upward. This suggests that most graupel production results from the freezing of raindrops, likely heterogeneously through interactions with pre-existing ice particles. Moreover, allowance of liquid supersaturation limits condensation in FSBM convective updrafts causing less LWC at T < -8 °C than in bulk schemes that condense all liquid supersaturation through

saturation adjustment while applying a constant cloud droplet number concentration. Although results are not presented here for brevity, the aerosol aware Thompson scheme (Thompson and Eidhammer, 2014) with explicit CCN activation had smaller raindrops than the non-aerosol aware Thompson scheme for -4 °C > T > 0 °C with supercooled drizzle drops rather than cloud droplets as in FSBM, supporting the hypothesized reasons for differences between FSBM and the bulk schemes.

Figures 11-14 show that condensate mass is partitioned differently among hydrometeor species for each scheme, which strongly impacts MD differences between schemes. MD differences also clearly rely on differences in the number of prognostic PSD moments for bulk schemes, mass-size relationships, PSD functions, fall-speed relationships, and microphysical process parameterizations. In particular, many studies have shown that hydrometeor sedimentation rates impact cloud and precipitation structure (e.g. Rutledge and Houze, 1987; Fovell and Ogurua, 1988; McCumber et al., 1991; Ferrier et al., 1995; Lynn et al., 2007; Thompson et al., 2008), and thus the assumed terminal velocity-size relationships described in Tables 1-3 may affect the model size biases presented. However, a sensitivity study of how variations in terminal velocity-size relationships impact MDs is beyond the scope of this study. Lastly, it is important to note that both graupel and snow MDs are larger than observed (cf. Fig. 6, 12, and 13), and therefore, overproduction of graupel in simulations is not solely responsible for the model hydrometeor size bias.

## 5.5 Connecting hydrometeor size biases to radar reflectivity biases

Equivalent Rayleigh reflectivity factor size distributions (ZSDs) describe how reflectivity is distributed by particle diameter. Simulated ZSDs are given by the integrands of Eq. (6) and Eq. (7) for ice and liquid, respectively, such that the combined-hydrometeor ZSD ($Z_e(D)_{tot}$), as with combined-hydrometeor MSDs (Eq. 4), are the combined-hydrometeor $Z_e$ distributions:

$$Z_e(D)_{tot} = 10^{18} \left[ 0.224 \sum_{i=1}^{n} \left( \frac{6\alpha_i}{\pi \rho_w} \right)^2 D^{2\beta_i} N_i(D) + \sum_{j=1}^{m} D^6 N_j(D) \right] \qquad (8)$$

where $n$ is the number of ice species and $m$ is the number of liquid species. Observed ZSDs for ice particles are calculated using the integrand of Eq. (6) where the diameter definition is the area-equivalent diameter ($D_{eq}$).

One caveat to interpreting differences between simulated and observed MSDs and ZSDs is that the observed MSDs for each 5-second flight sample are characterized by a single retrieved mass-diameter power law, which may not work well in all situations, for example when relatively high density small and large particles co-exist with low density medium-sized particles. However, retrieval bias is likely limited by comparison of composite distributions, which are computed by taking the mean mass for the composite MSD and mean equivalent reflectivity factor for the composite ZSD in each particle diameter bin. Furthermore, observed particles counts are more uncertain in the distribution tails at larger diameters (i.e. diameters > 3 mm). There is greater confidence placed in observed particle count, mass, and density for diameters smaller than 3 mm, by which an order of magnitude difference between observed and simulated masses or reflectivities at a given particle diameter requires greater than a 300% particle density error for the ZSD and 1000% particle density error for the MSD.

Figure 15 shows observed and simulated composite PSDs, MSDs, and ZSDs for T between -32 °C and -40 °C. All observed and simulated data points where TWC is between 2 and 2.5 g m$^{-3}$ are included so that each composite distribution has approximately the same bulk mass. Simulated TWC constraints are limited to the observed diameter spectrum (maximum of 12.85 mm) for consistency among integrated distributions. Note that observed PSDs and MSDs include both $D_{eq}$ (black) and $D_{max}$ (grey) particle size definitions. All simulations struggle to reproduce the observed PSD using both the $D_{eq}$ and $D_{max}$

definitions with less particles than observed at diameters smaller than 0.5 mm. The Thompson scheme reproduces the observed profile reasonably well for diameters larger than 0.5 mm, while FSBM produces an order of magnitude more large particles than observed for diameters larger than 3 mm. Morrison produces the wrong PSD slope, distributing too many particles at sizes between 1 and 5 mm, but less particles than observed outside this range.

5        Composite MSDs (Fig. 15b) include symbols indicating MMDs (asterisks) and 90% MDs (triangles). Observations using the $D_{eq}$ and $D_{max}$ size definitions are relatively similar for sizes up to ~ 2.5 mm, but less mass is distributed at larger $D_{max}$ values than at larger $D_{eq}$ values. The observed MSD shows a prominent particle mass mode at ~ 300 µm, and remarkably, 90% of the > 2 g m$^{-3}$ condensate mass observed is typically contained in particles with diameters < 1 mm. Thompson is the only simulation able to reproduce the prominent mass mode, whereas the FSBM and Morrison schemes shift it to larger sizes.

Thompson reproduces observed mass for sizes up to 1.5 mm, but diverges significantly from observations at larger sizes, with differences as large as an order of magnitude for sizes between 2 and 5 mm. This is largely a result of the large graupel sizes produced by this scheme (see Fig. 13). FSBM captures the shape of the observed MSD reasonably well, but distributes too much mass at particles larger than ~ 500 µm, and too little mass at smaller diameters. The Morrison scheme again produces the wrong distribution slope, producing the largest discrepancies with observations for sizes between 1 and 4 mm. Simulated

MMDs are closer to observed 90% MDs than observed MMDs, whereas simulated 90% MDs are several millimeters larger than observed. As noted in Sect. 5.4, this bias is not eliminated by considering simulated snow alone because both snow and graupel MDs are larger than observed MDs that consider all ice types.

        Composite ZSDs for the $D_{eq}$ definition alone are shown in Fig. 15c. The observed composite ZSD produces a reflectivity peak mode between 0.5 and 1 mm and a local minimum between 2 and 4 mm with a secondary mode at larger

diameters. Both bulk schemes fail to capture the local minimum around 2-4 mm and produce larger reflectivities than retrieved between 2 and 5 mm where differences exceed an order of magnitude. FSBM also produces larger $Z_e$ values compared to observations, despite being the only scheme that captures the local ZSD minimum. Notably, Thompson reproduces the observed ZSD up to 1.5 mm similarly to the MSD, but produces much higher reflectivities at larger diameters despite having only 10% of condensate mass distributed in these larger diameters. Recall that particle density errors of 1-2 orders of magnitude would be required to produce these large differences between observed and simulated $Z_e$, and thus there is confidence that

these discrepancies are not entirely attributable to sampling bias, particularly for sizes smaller than 3 mm. It is also worth noting that comparison of median MSDs and ZSDs (not shown) shows that a significant fraction of observed distributions have little to no particles larger than a few millimeters, while this is rarely, if ever, produced in simulations for TWC > 2 g m$^{-3}$.

Composite distributions were additionally examined for a warmer temperature range (-8 °C to -20 °C), and differences between observations and simulations are still present and significant, although observational sample sizes are much smaller at these temperatures. Notably, no scheme in the warmer T range is able to capture the prominent mass peak at ~ 300 µm that persists in observations, even at these warmer temperatures. This includes the Thompson scheme, suggesting that Thompson's ability to reproduce this mass mode at colder T, where SWC is the dominant bulk mass species, is due to the snow PSD

parameterization that diagnostically forces particles to smaller sizes at colder T rather than more realistic physical process parameterizations. Overall, analysis of these ZSDs show that even though simulations often only distribute 10% of condensate mass at diameters larger than 2-3 mm, that small fraction of mass is often much greater than observed and greatly biases radar reflectivity.

## 6 Conclusions

Properly representing cloud microphysical processes and hydrometeor properties in microphysics parameterizations is vital to improving simulations of clouds and precipitation, but identifying sources of model bias is difficult given the complexity of nonlinear interactions between dynamics and microphysics within the model. This study differs from previous studies by controlling for vertical velocity, bulk condensate mass, and temperature to isolate the contribution of hydrometeor size from the contribution of excessive condensate mass in overly strong or large convective updrafts to the well-known high bias in simulated tropical deep convective radar reflectivity.

Data collected during the first HAIC-HIWC field campaign held in Darwin, Australia, in 2014, are compared with three WRF simulations of a mesoscale convective system that passed through Darwin on 18 February. The simulations vary only by the microphysics scheme employed (Thompson, Morrison, and FSBM). While a reflectivity bias in simulated tropical convection has been previously shown to exist in bulk schemes, this study shows that the bias exists in a bin scheme (FSBM) as well. Simulated MMDs and upper percentile MDs are larger than observed in every microphysics scheme for a given $w$, TWC, and T condition. Many TWC-T and $w$-T conditions exist in each scheme where the minimum 90% MD is larger than observed, despite simulated samples sizes being ~ $10^2$ times larger than observations.

For temperatures between -30 °C and -50 °C, the Thompson scheme best reproduces observed MMDs for a given TWC or $w$. Vapor-grown ice particles largely control the bulk mass at these temperatures, and the Thompson scheme uses a unique snow PSD parameterization combined with a $m$-$D$ relationship that forces snow mass into smaller sizes with decreasing T. Although these small, dense snow particles are diagnosed rather than produced by a microphysical process, they suggest that using a $m$-$D$ relationship in which vapor-grown ice particle density decreases with particle size combined with an appropriate PSD function can nearly reproduce observations at cold temperatures where vapor-grown ice contributes most to condensate mass. However, the Thompson scheme produces larger than observed MMDs for temperatures between -10 °C and -30 °C, especially for high TWC or $w$, where graupel largely controls condensate mass. The graupel size is diagnostically increased as graupel bulk mass increases, pushing graupel to large sizes. While large graupel particles have a hail-like fall-speed that sediments them out of updrafts more quickly than in the Morrison and FSBM schemes, their sizes become so large that only a small graupel bulk mass easily high biases radar reflectivity.

The Morrison scheme allows for much greater variability in graupel and snow size since it predicts $N$ for both species, making it a two-moment ice scheme. This shifts graupel to smaller mean sizes and snow to larger mean sizes than in the Thompson scheme. Indeed, snow MMDs are on average larger than graupel MMDs in the Morrison scheme, and both are larger than observed MMDs for most T-$w$-TWC conditions. However, fall-speeds of Morrison graupel that are significantly

slower than Thompson graupel fall speeds result in much of the bulk mass being controlled by GWC in updraft cores, even at temperatures down to -50 °C. This causes the combined-hydrometeor MMDs in Morrison to be largely controlled by graupel.

The FSBM scheme has a fundamental advantage over the bulk schemes in that it does not assume a PSD shape and computes microphysical process rates separately for different hydrometeor size bins. Similarly to the Morrison scheme, snow MMDs are larger than graupel MMDs in the FSBM scheme, which combined with large amounts of SWC, indicate that snow is largely responsible for size biases in FSBM. Notably, graupel sizes in FSBM generally remain below 1 mm for most T-*w*-TWC bins. These smaller graupel sizes may result from two processes unique to the FSBM scheme: explicit CCN nucleation and maintenance of liquid supersaturation. These processes may aid in reducing the size of lofted raindrops that freeze upon collision with ice particles to form most graupel, while less supercooled liquid limits additional riming.

Perhaps most revealing are features from composite mass and equivalent Rayleigh reflectivity factor distributions as a function of diameter. A prominent mass mode at ~ 300 µm exists in observed MSDs regardless of T, *w*, or TWC constraints, but rarely are any of the schemes able to reproduce this feature, with the exception of the Thompson scheme at temperatures colder than -30 °C. Otherwise, all schemes produce too much mass at large particle diameters, even for TWC < 1 g m$^{-3}$, although discrepancies with observations are enhanced for larger TWCs and higher *w* values. The excess simulated mass at diameters > 1 mm leads to reflectivity factors that are higher than observed by up to two orders of magnitude in some diameter ranges.

Ultimately, all simulations fail to reproduce observed hydrometeor size distributions in which the majority of bulk mass is distributed at sub-mm sizes like commonly observed. Bulk scheme mass distributions are sensitive to assumed hydrometeor properties including the PSD function and the mass-size relationship. The bin scheme failures show that additional causes of hydrometeor size biases are likely related to species partitioning and parameterization of microphysical processes. Biases resulting from microphysical processes are likely present in bulk schemes as well, but further research is needed to determine how much of the bias results from microphysical process parameterization errors versus diagnosed single particle and PSD properties. Future work using data from the second HAIC-HIWC phase held in Cayenne, French Guiana, in 2015 will enable more in-depth evaluation of biases in deep convection at temperatures warmer than -15 °C, where size biases appear to originate in convective updrafts.

*Competing interests.* The authors declare that they have no conflict of interest.

*Data availability.* Simulation output is available upon request by the correspondence author. C-POL data for the HAIC-HIWC time period is available upon request through the Bureau of Meteorology (point of contact: Alain Protat, alain.protat@bom.gov.au). In accordance with HAIC-HIWC data sharing protocol, SAFIRE, MSD, and IKP2 data will be made conditionally available to the public on 15 July 2019 and without conditions on 15 July 2022. Conditional availability requires agreement of the HAIC-HIWC data sharing protocol and conditions regarding co-authorship.

*Acknowledgements.* This research was supported by the National Science Foundation (NSF) award number 1213310 with computing resources and support provided by the Center for High Performance Computing at the University of Utah. Special thanks are given to the Bureau of Meteorology (BoM) for providing meteorological products and C-POL data and SAFIRE for providing aircraft and environmental state data. The Japan Meteorological Agency is thanked for the provision of real-time rapid-scan visible and infrared satellite data from MTSAT-1R during the Darwin campaign. We thank NCAR EOL under sponsorship of NSF, the BoM, and Laboratoire de Météorologie Physique (LaMP) for data collection, storage, and dissemination. Experiment and data processing funding was provided by the European Union's Seventh Framework Program in research, technological development, and demonstration under grant agreement n°ACP2-GA-2012-314314, the EASA Research Program under service contract n° EASA.2013.FC27, and the FAA Aviation Research and Weather Divisions under agreement CON-I-1301 with the Centre National de la Recherche Scientifique. Additional support was provided by the NASA Aviation Safety Program, the Boeing Co., Transport Canada, Airbus Operations SAS, Science Engineering Associates, the BoM, Environment Canada, the National Research Council of Canada, and the NSF under grant AGS 12-13311. Thanks are also given to Dr. Steven Utembe at the University of Melbourne for providing code to produce WRF forcing files from ACCESS-R analyses.

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

**Table 1**. Thompson mass-size relationship ($m = \alpha D^\beta$), terminal velocity-size relationship ($v = [\rho_0/\rho]^{1/2} c D^d exp[-fD]$), and gamma PSD parameter values used for each species. $N_c$ used for cloud water $\mu$ and $N_0$ calculations is a constant 100 cm$^{-3}$. $N_0$ equations for double-moment species are determined by prognostic $N$ and $q$. The bimodal gamma distribution used for the Thompson snow PSD may be found in Thompson et al. (2008), Eq. (1). The $f$ parameter in the terminal velocity-size relationship follows Ferrier (1994).

| | | | | | | | | Thompson $m$-$D$, $v$-$D$, and PSD parameters | |
|---|---|---|---|---|---|---|---|---|---|
| Species | Prognostic Variables | $\rho$ [kg m$^{-3}$] | $\alpha$ | $\beta$ | $c$ | $d$ | $f$ | $N_0$ [m$^{-4}$] | $\mu$ |
| Snow | $q_s$ | $\frac{6\alpha_s}{\pi} D^{\beta_s-3}$ | 0.069 | 2 | 40 | 0.55 | 100 | - | 0.6357 |
| Graupel | $q_g$ | 500 | $\frac{\pi \rho_g}{6}$ | 3 | 442 | 0.89 | 0 | $max\left[10^4, min\left(\frac{200}{q_g}, 3\times10^6\right)\right]$ | 0 |
| Cloud Ice | $q_i$, $N_i$ | 890 | $\frac{\pi \rho_i}{6}$ | 3 | 1847.5 | 1 | 0 | $\frac{N\lambda^{\mu+1}}{\Gamma(\mu+1)}$ | 0 |
| Rain | $q_r$, $N_r$ | 1000 | $\frac{\pi \rho_w}{6}$ | 3 | 4854 | 1 | 195 | $\frac{N\lambda^{\mu+1}}{\Gamma(\mu+1)}$ | 0 |
| Cloud Water | $q_c$ | 1000 | $\frac{\pi \rho_w}{6}$ | 3 | $0.316946 \times10^8$ | 2 | - | $\frac{N_c\lambda^{\mu+1}}{\Gamma(\mu+1)}$ | $min(15, \frac{10^9}{N_c} + 2)$ |

**Table 2**. Morrison mass-size relationship ($m = \alpha D^\beta$), terminal velocity-size relationship ($v = [\rho_0/\rho]^{0.54} cD^d$), and gamma PSD parameter values used for each species. $N_c$ used for cloud water $\mu$ and $N_0$ calculations is a constant 100 cm$^{-3}$. Cloud water $\mu$ is calculated as a function of $N_c$ using an empirical relationship described in Martin et al. (1994).

| | | Morrison $m$-$D$, $v$-$D$, and PSD parameters | | | | | | |
|---|---|---|---|---|---|---|---|---|
| Species | Prognostic Variables | $\rho$ [kg m$^{-3}$] | $\alpha$ | $\beta$ | $c$ | $d$ | $N_0$ [m$^{-4}$] | $\mu$ |
| Snow | $q_s$, $N_s$ | 100 | $\dfrac{\pi \rho_s}{6}$ | 3 | 11.72 | 0.41 | $\dfrac{N\lambda^{\mu+1}}{\Gamma(\mu+1)}$ | 0 |
| Graupel | $q_g$, $N_g$ | 400 | $\dfrac{\pi \rho_g}{6}$ | 3 | 19.3 | 0.37 | $\dfrac{N\lambda^{\mu+1}}{\Gamma(\mu+1)}$ | 0 |
| Cloud Ice | $q_i$, $N_i$ | 500 | $\dfrac{\pi \rho_i}{6}$ | 3 | 700 | 1 | $\dfrac{N\lambda^{\mu+1}}{\Gamma(\mu+1)}$ | 0 |
| Rain | $q_r$, $N_r$ | 997 | $\dfrac{\pi \rho_w}{6}$ | 3 | 841.99667 | 0.8 | $\dfrac{N\lambda^{\mu+1}}{\Gamma(\mu+1)}$ | 0 |
| Cloud Water | $q_c$ | 997 | $\dfrac{\pi \rho_w}{6}$ | 3 | $3\times10^7$ | 2 | $\dfrac{N_c\lambda^{\mu+1}}{\Gamma(\mu+1)}$ | - |

**Table 3**. FSBM mass-size relationship ($m = \alpha D^\beta$) parameters and terminal velocity ranges as a function of size ($v$-$D$). Note that the snow density varies by bin and thus a range is given. Ranges are also given for all $v$-$D$ relationships. The subscript $k$ for prognostic variables refers to the $k^{th}$ size bin for a given species.

| | | FSBM $m$-$D$ parameters $v$-$D$ relationships | | | |
|---|---|---|---|---|---|
| Species | Prognostic Variables | $\rho$ [kg m$^{-3}$] | $\alpha$ | $\beta$ | $v$-$D$ |
| Snow | $q_{s,k}$, $N_{s,k}$ | Decreases from 900 kg m$^{-3}$ at $D$=4.14 µm to 35 kg m$^{-3}$ at $D$=19.87 mm | $\dfrac{\pi \rho_{s,k}}{6}$ | 3 | Increases from $2 \times 10^{-4}$ m s$^{-1}$ at $D$=4.14 µm to 1.4 m s$^{-1}$ at $D$=19.87 mm |
| Graupel | $q_{g,k}$, $N_{g,k}$ | 400 | $\dfrac{\pi \rho_g}{6}$ | 3 | Increases from $3.9 \times 10^{-4}$ m s$^{-1}$ at $D$=5.43 µm to 4 m s$^{-1}$ at $D$=8.82 mm |
| Liquid | $q_{r,k}$, $N_{r,k}$ | 1000 | $\dfrac{\pi \rho_w}{6}$ | 3 | Increases from $5 \times 10^{-4}$ m s$^{-1}$ at $D$=4 µm to 9 m s$^{-1}$ at $D$=6.5 mm |

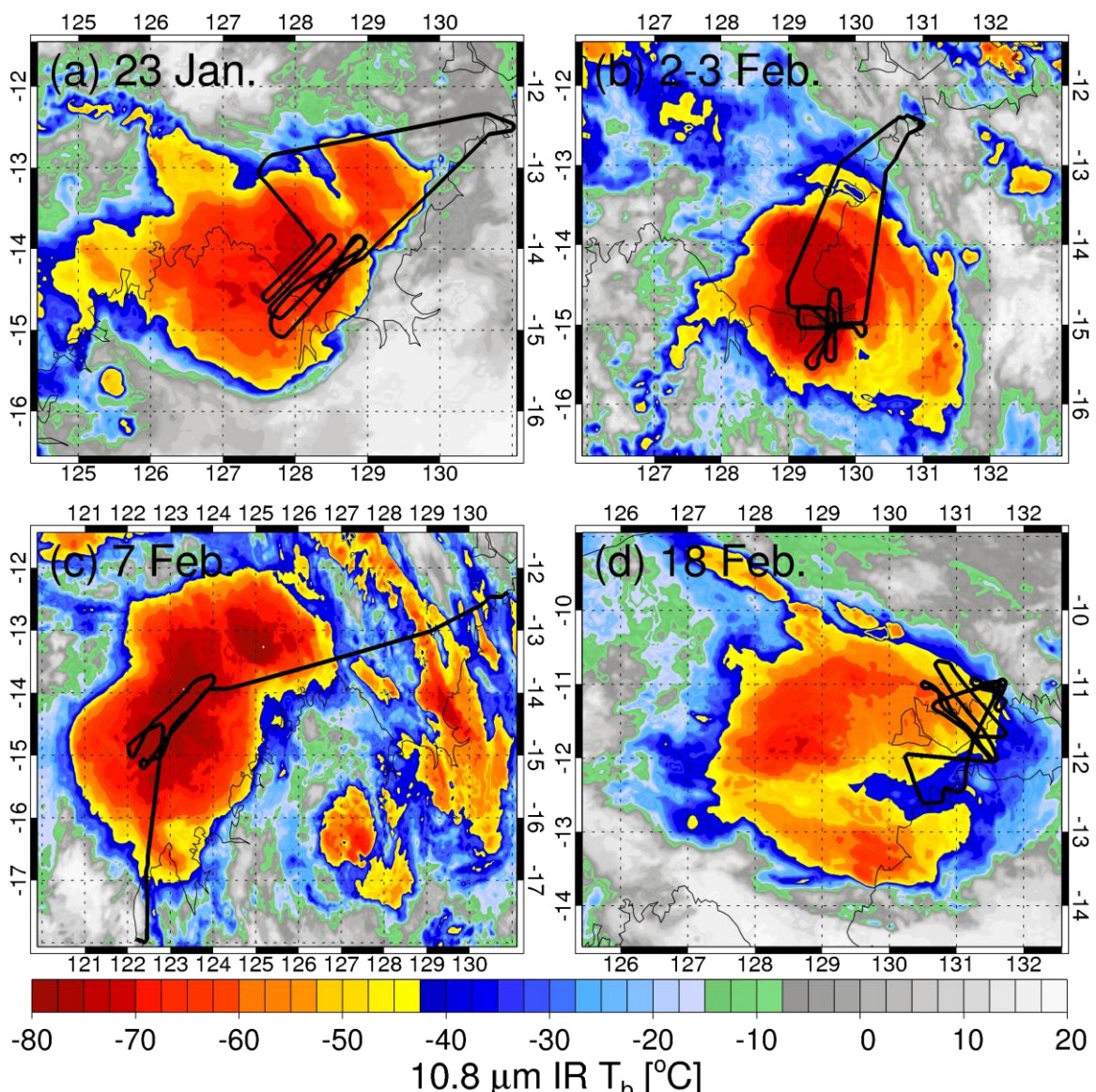

**Figure 1**: Flight tracks (black lines) for (a) Flight 6 on 23 January 2014, (b) Flight 13 on 3 February 2014, (c) Flight 16 on 7 February 2014, and (d) Flight 23 on 18 February 2014 overlaid on IR imagery from MTSAT-1R during flight sampling. Note that both Flights 12 and 13 flew through the same system (2-3 February), but only Flight 13 is shown.

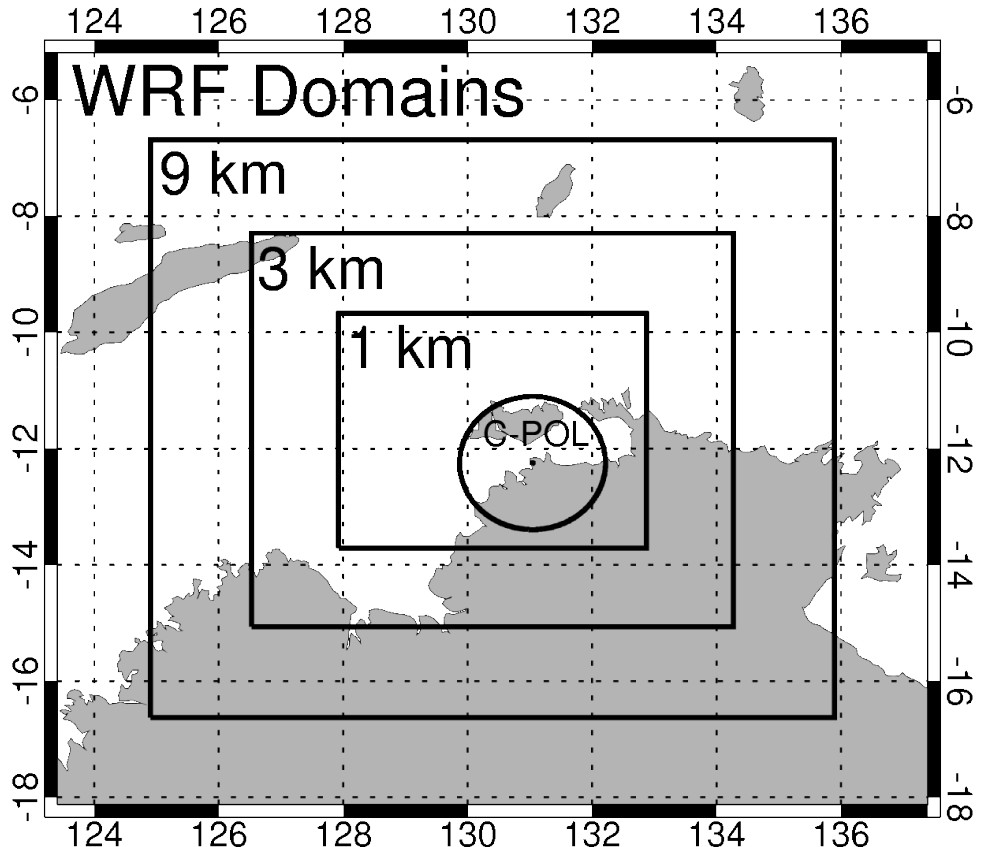

**Figure 2.** The WRF domains used for the 18 February 2014 simulation. The circle indicates the 150-km range ring of the C-POL radar located at Gunn Point.

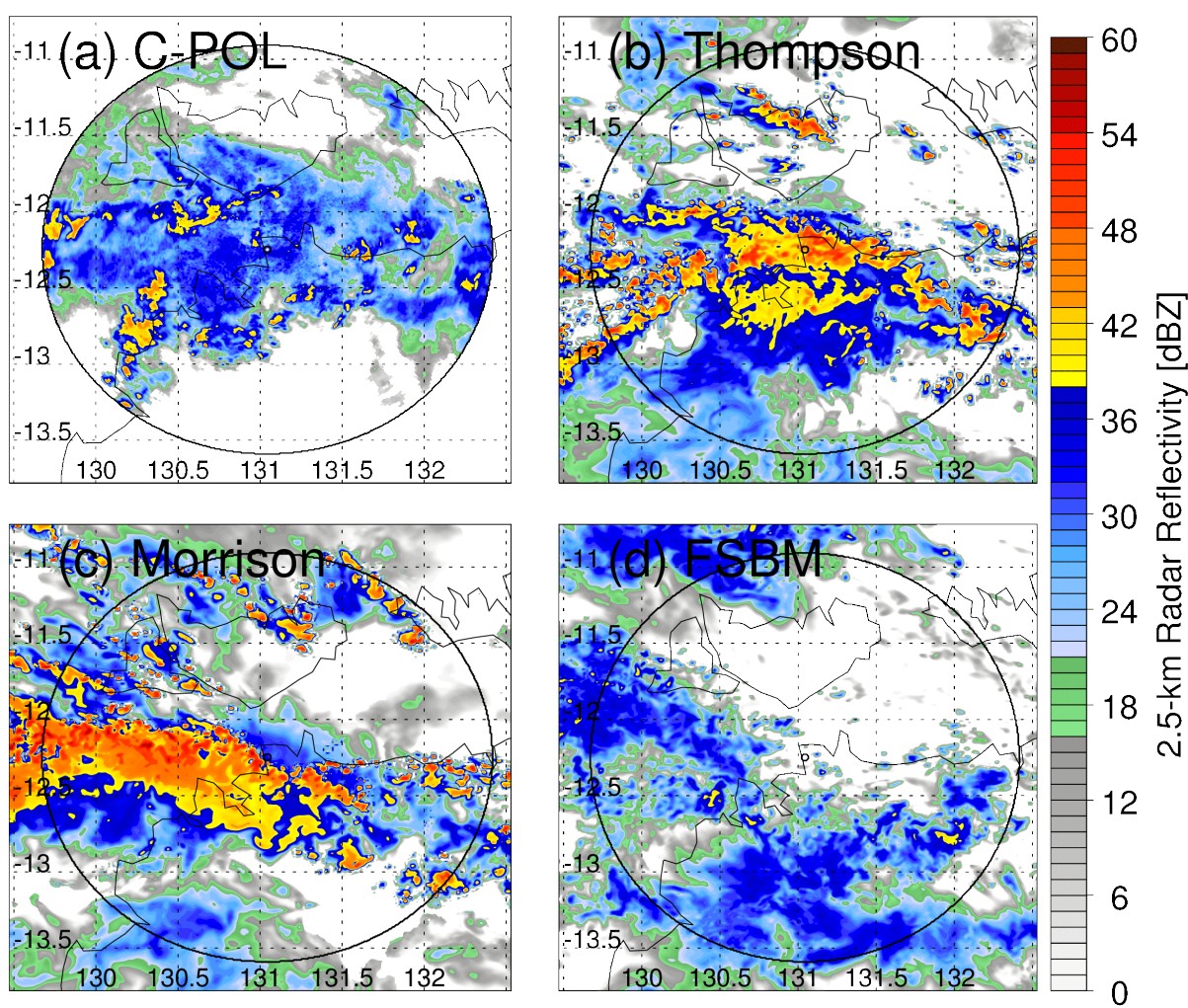

**Figure 3.** Representative 2.5-km altitude horizontal radar reflectivity cross-sections on 18 February 2014 at 18Z for (a) C-POL observations, (b) Thompson, (c) Morrison, and (d) FSBM schemes. The circle indicates the 150-km range ring of the C-POL radar.

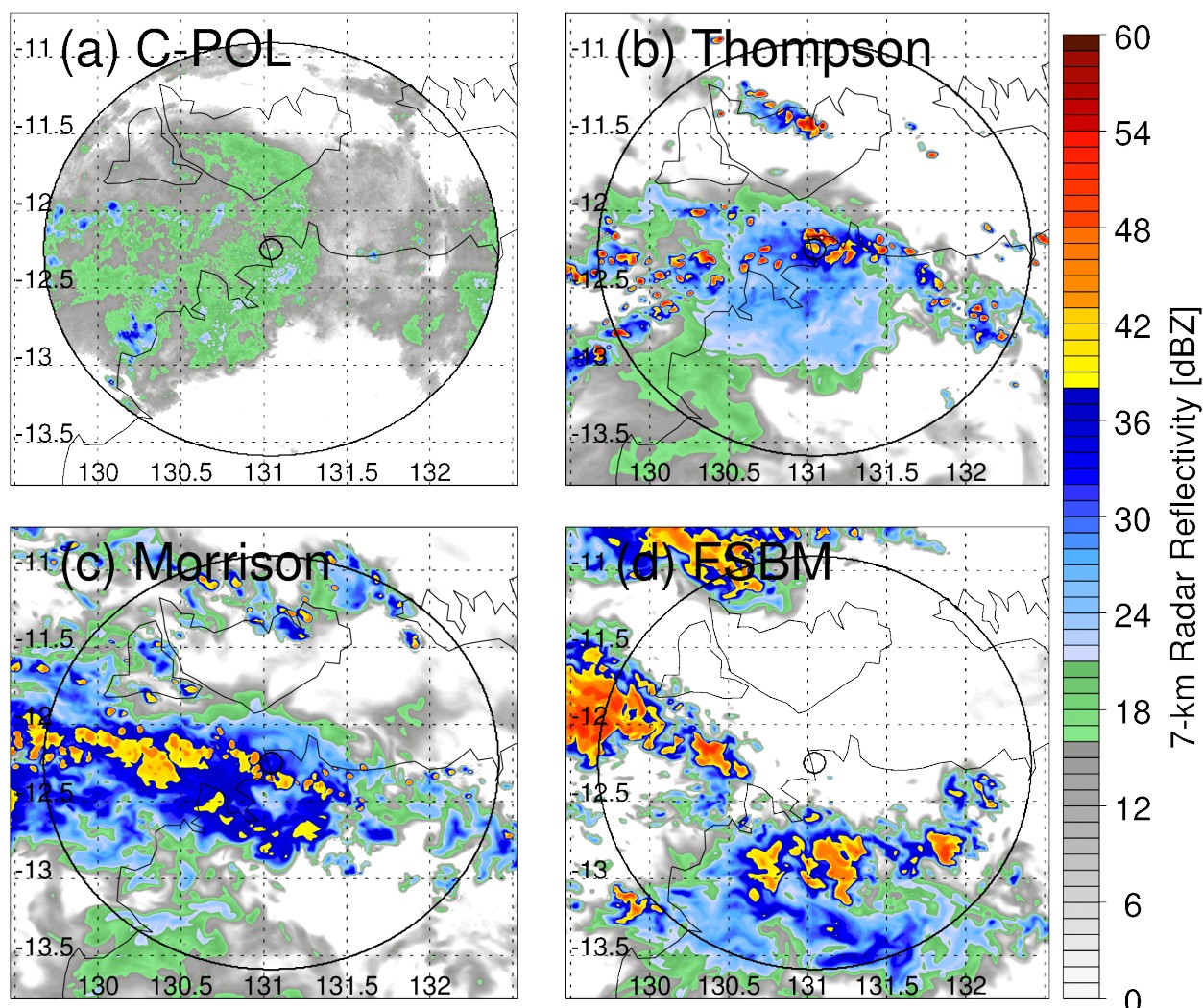

**Figure 4.** As in Fig. 3, but for 7-km altitude.

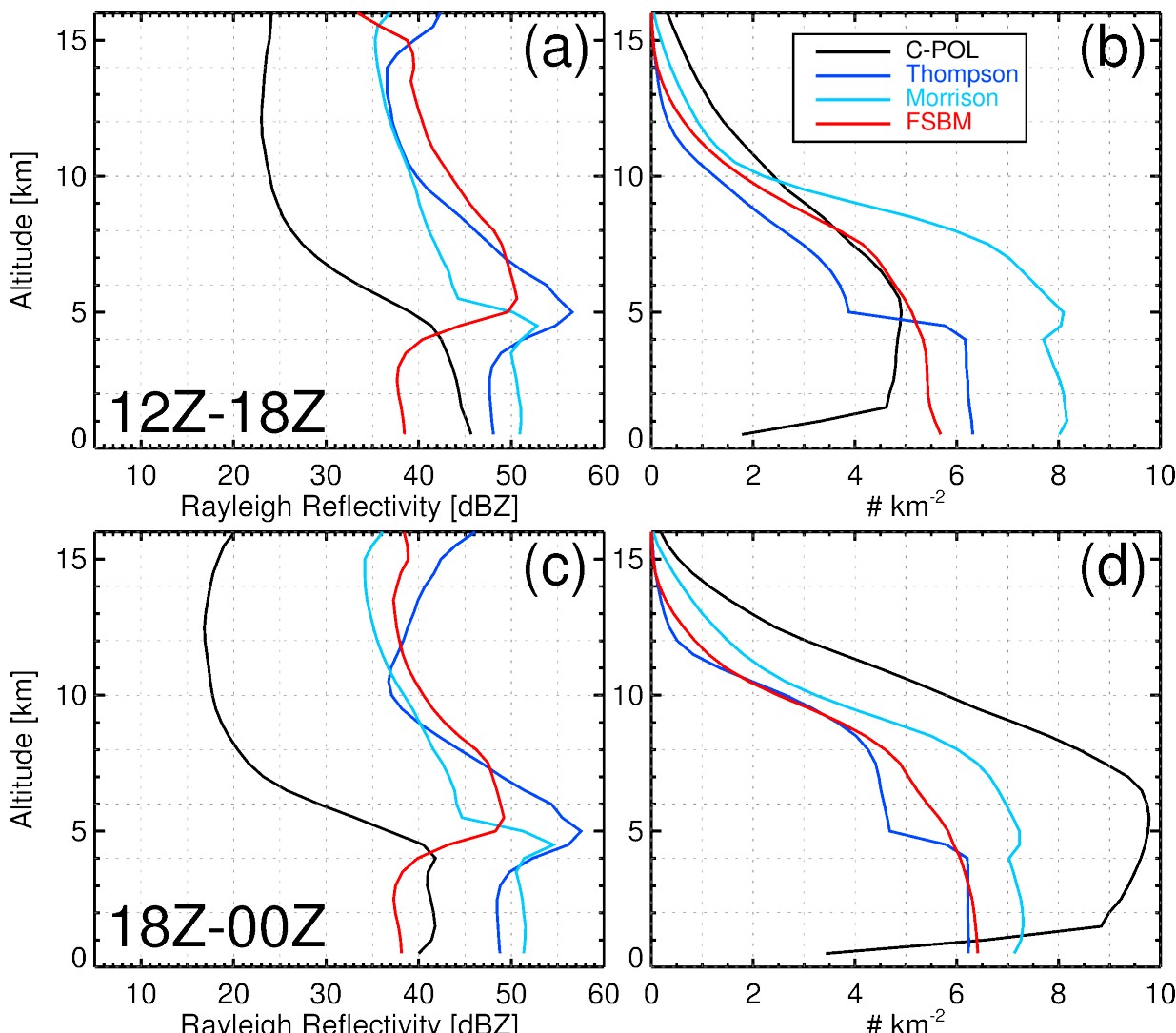

**Figure 5.** Rayleigh reflectivity 99[th] percentile profiles for the observed and simulated 18 February 2014 MCS case during (a) 12Z-18Z on the 18[th] and (c) 18Z on the 18[th] to 00Z on the 19[th]. Sample sizes normalized by domain area are shown in (b) and (d) for 12Z-18Z and 18Z-00Z, respectively. Only reflectivity values ≥ 5 dBZ are included. Retrieved C-POL reflectivity is in black, Thompson in dark blue, Morrison in cyan, and FSBM in red.

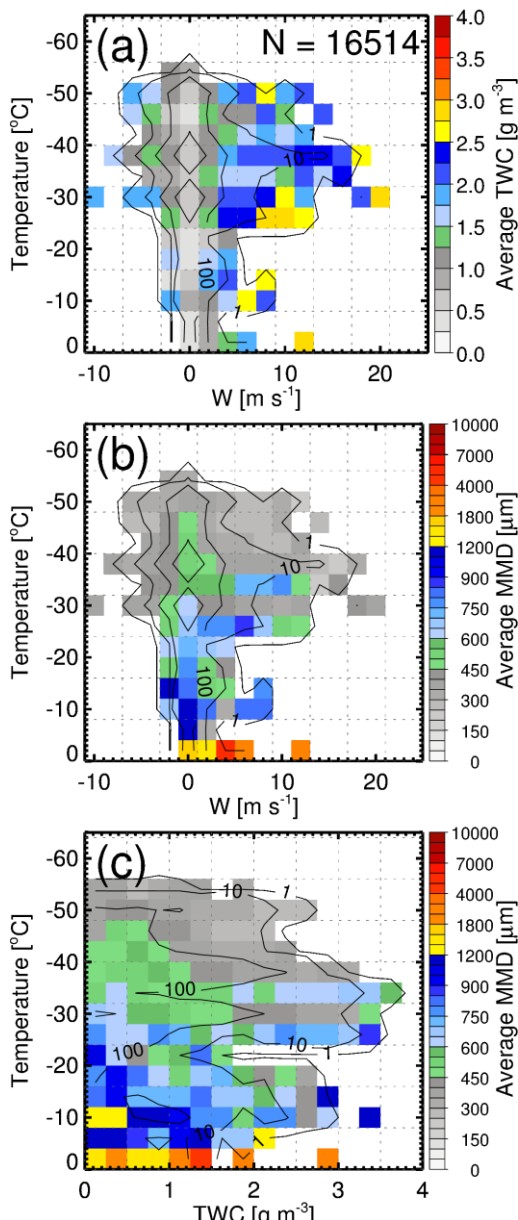

**Figure 6.** Observed joint histograms using Darwin HAIC-HIWC data: (a) w-T bins color-filled with average TWC, (b) w-T bins color-filled with average MMD, and (c) TWC-T bins color-filled with average MMD. Observational sample size is shown in the top right corner and order of magnitude sample sizes are contoured in black. Bin sizes are 2 m s⁻¹ for *w*, 0.25 g m⁻³ for TWC, and 4 °C for temperature. Note that

5     the average MMD color-scale is nonlinear and accentuates variability for average MMDs < 1 mm.

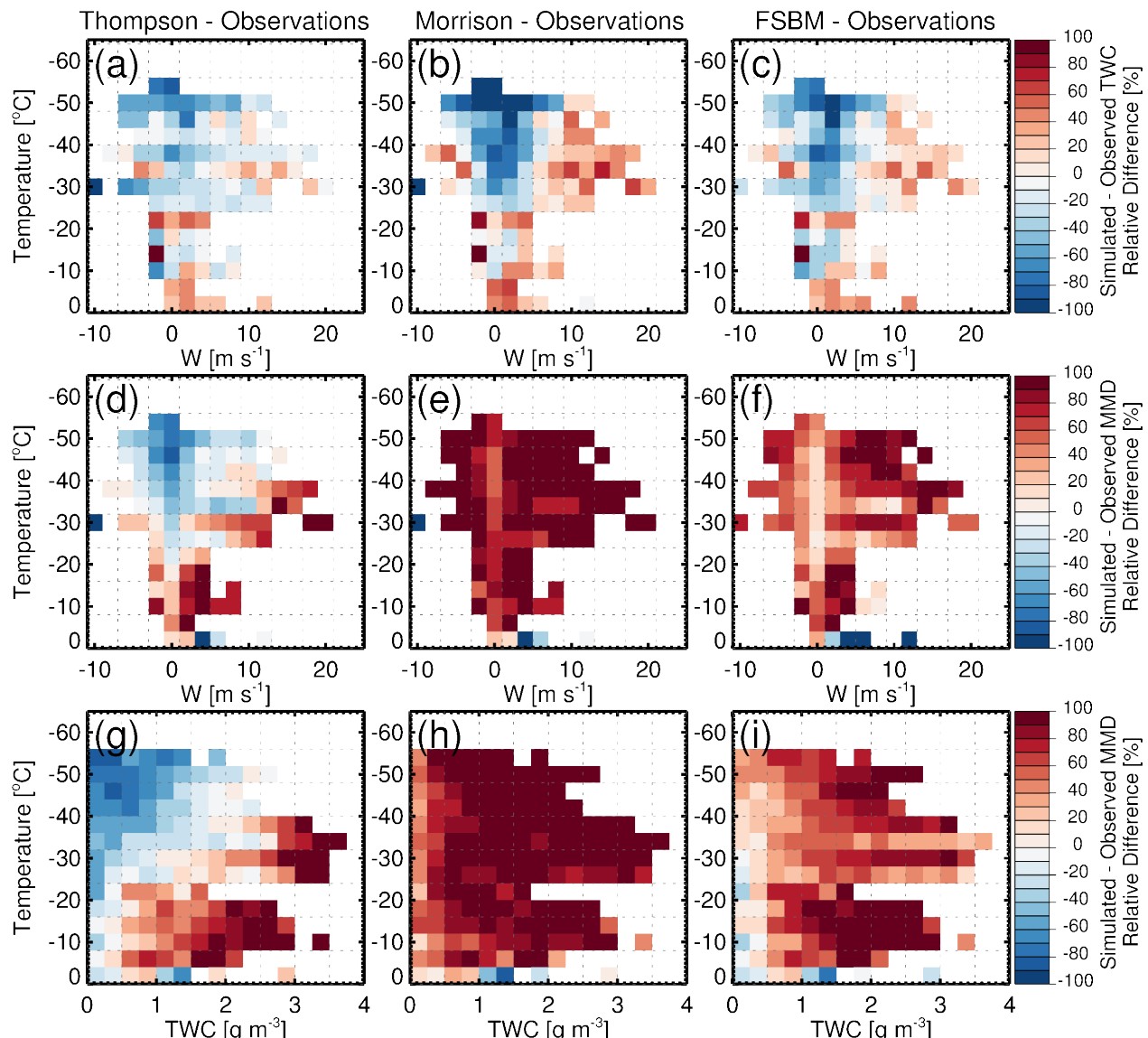

**Figure 7.** Joint histograms color-filled with relative differences between observed and simulated (a)-(c) TWC as a function of *w*-T bins, (d)-(f) MMD as a function of *w*-T bins, and (g)-(i) MMD as a function of TWC-T bins. Differences for the Thompson scheme are shown in the left column, the Morrison scheme in middle column, and the FSBM scheme in right column. Observed values are subtracted from simulated values, and only bins where observational data are available are shown.

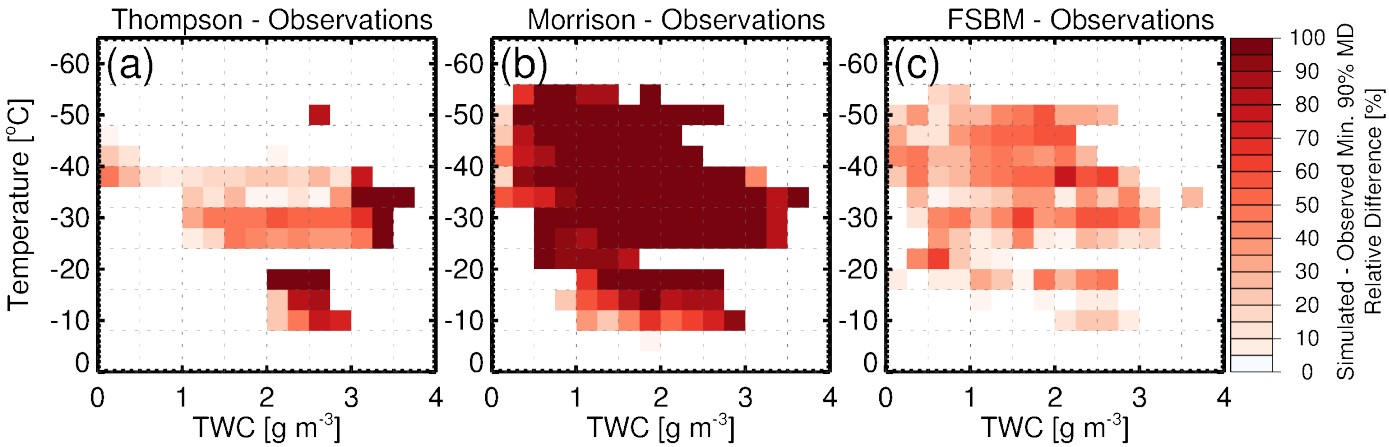

**Figure 8.** Joint histograms of positive relative differences between observed and simulated minimum 90% MDs as a function of TWC-T bins for (a) Thompson, (b) Morrison, and (c) FSBM schemes.

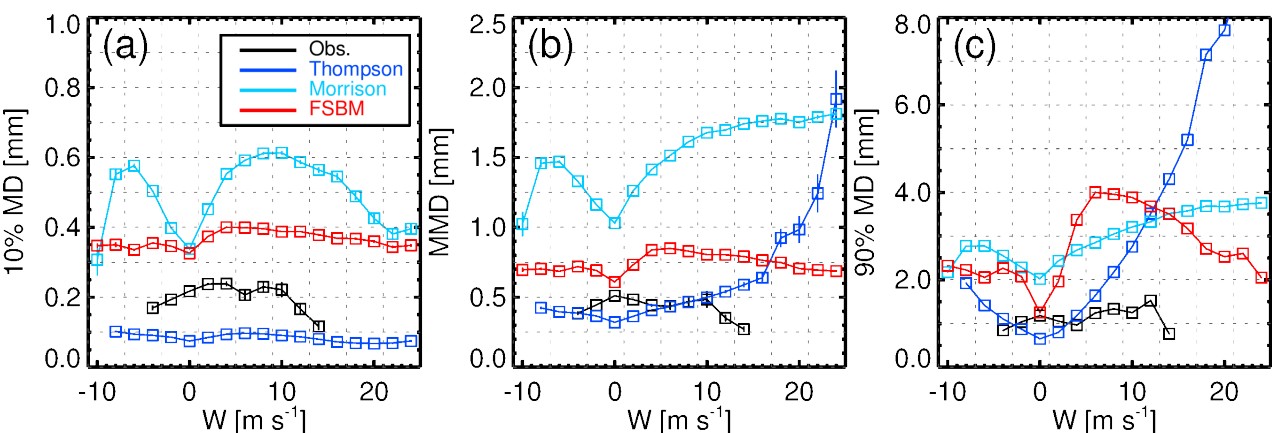

**Figure 9.** Average percentile mass diameters as a function of w bins (2 m s$^{-1}$ bin widths) for temperatures between -32 °C and -40 °C. 10% MDs are shown in (a), MMDs in (b), and 90% MDs in (c) with observations in black, Thompson in dark blue, Morrison in cyan, and FSBM in red. Vertical bars extending from each box represent standard errors; however, note that standard errors are rather small due to relatively large sample sizes.

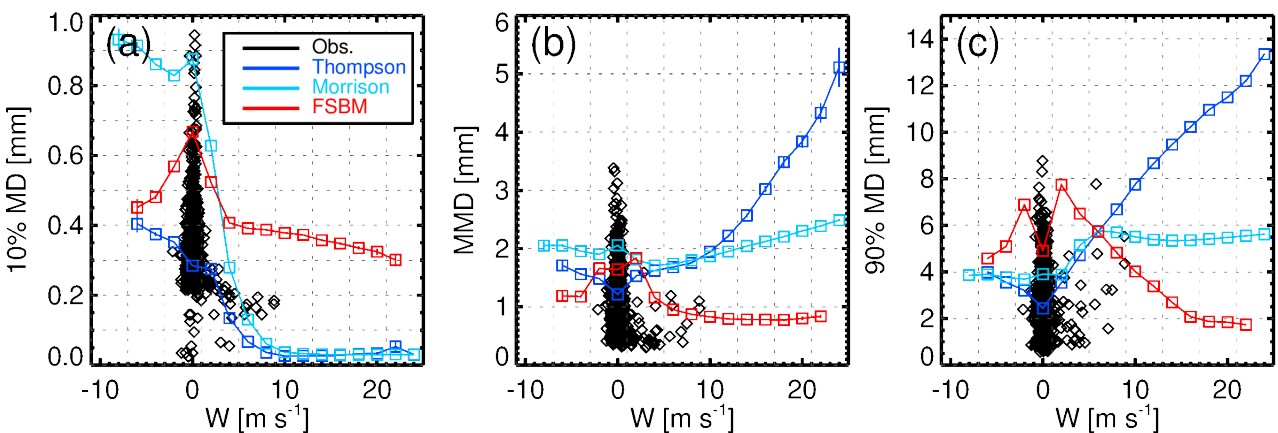

**Figure 10.** As in Fig. 11, but for temperatures between -8 °C and -16 °C. Observations are plotted as black diamonds displaying individual data points due to lower sample sizes in this temperature range.

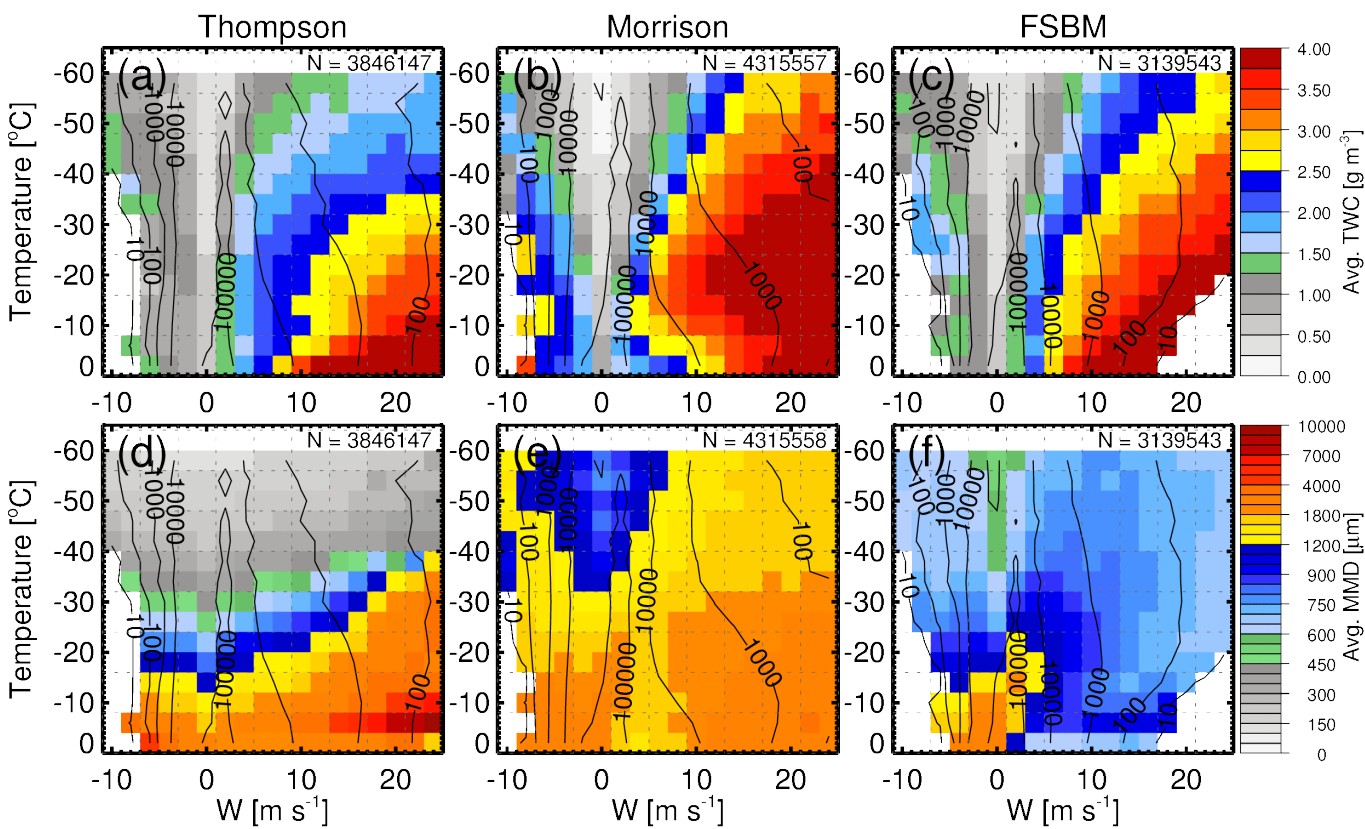

**Figure 11.** Joint histograms of w and temperature below freezing using bins sizes of 2 m s$^{-1}$ for w and 4 °C for temperature. Color-fill is average TWC for (a)-(c) and average combined-hydrometeor MMD for (d)-(f). The Thompson scheme is shown in (a) and (d), the Morrison scheme in (b) and (e), and the FSBM scheme in (c) and (f). Sample sizes are shown in the upper right corner, and order of magnitude sample sizes are contoured in black.

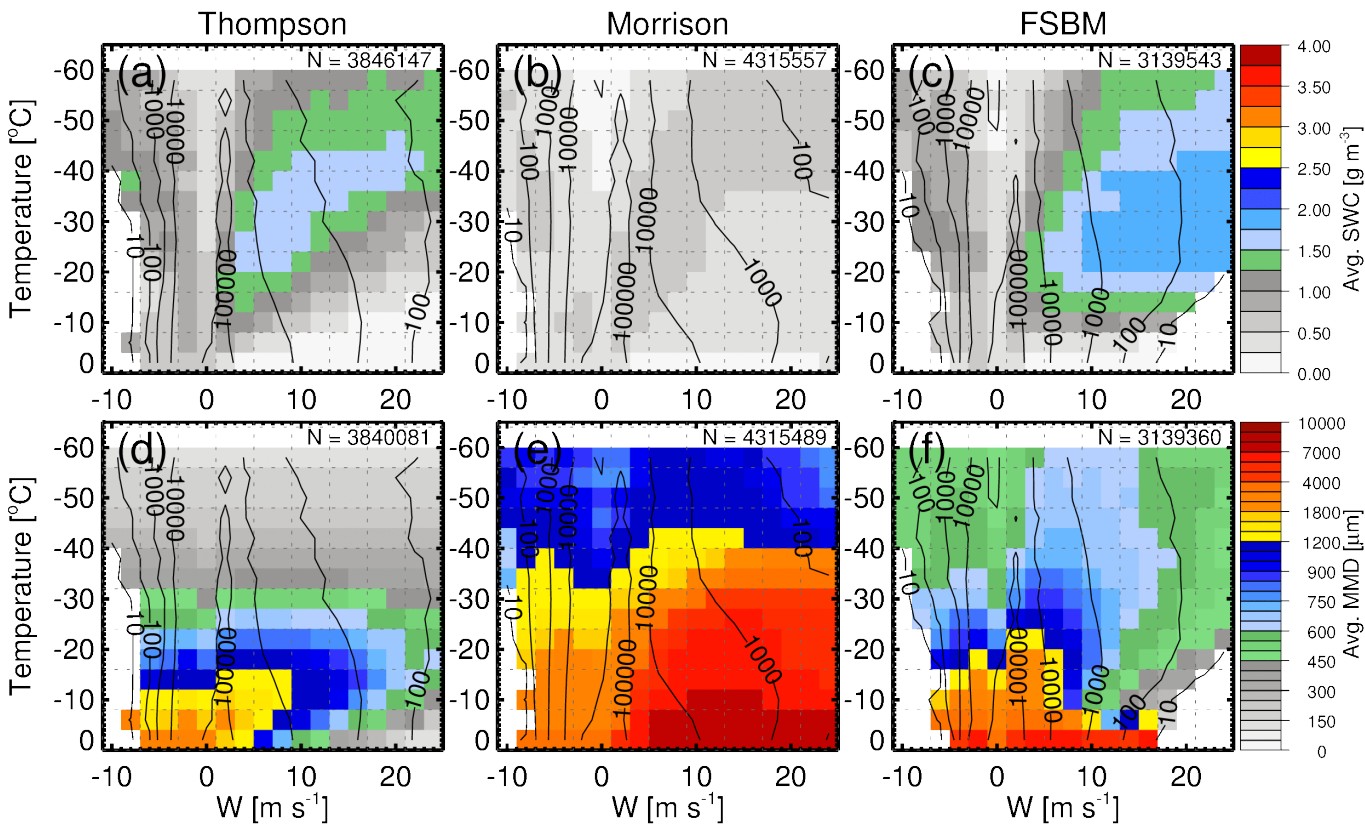

**Figure 12.** As in Fig. 11, but for snow water content (SWC) color-filled in (a)-(c) and average snow MMDs color-filled in (d)-(f).

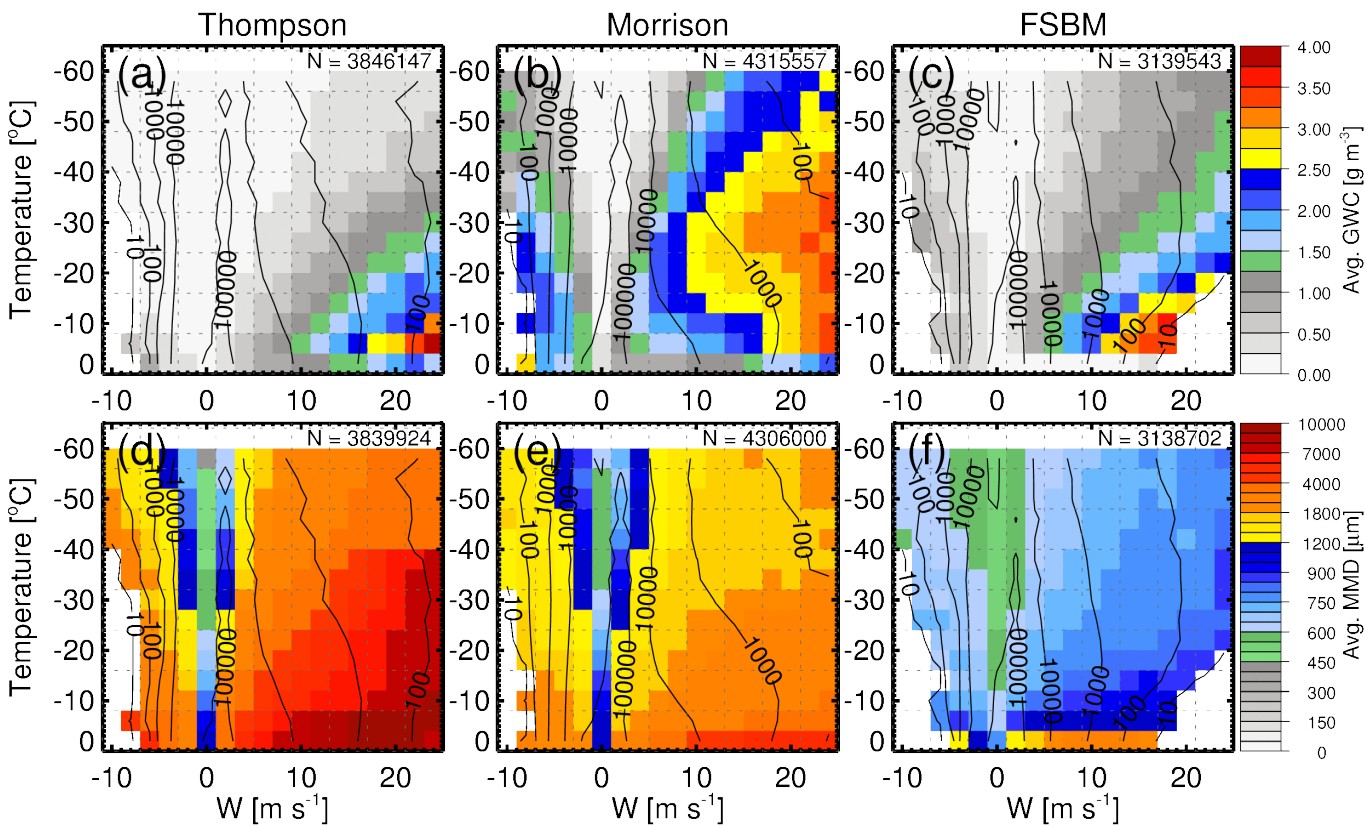

**Figure 13.** As in Fig. 11, but for graupel water content (GWC) color-filled in (a)-(c) and average graupel MMDs color-filled in (d)-(f).

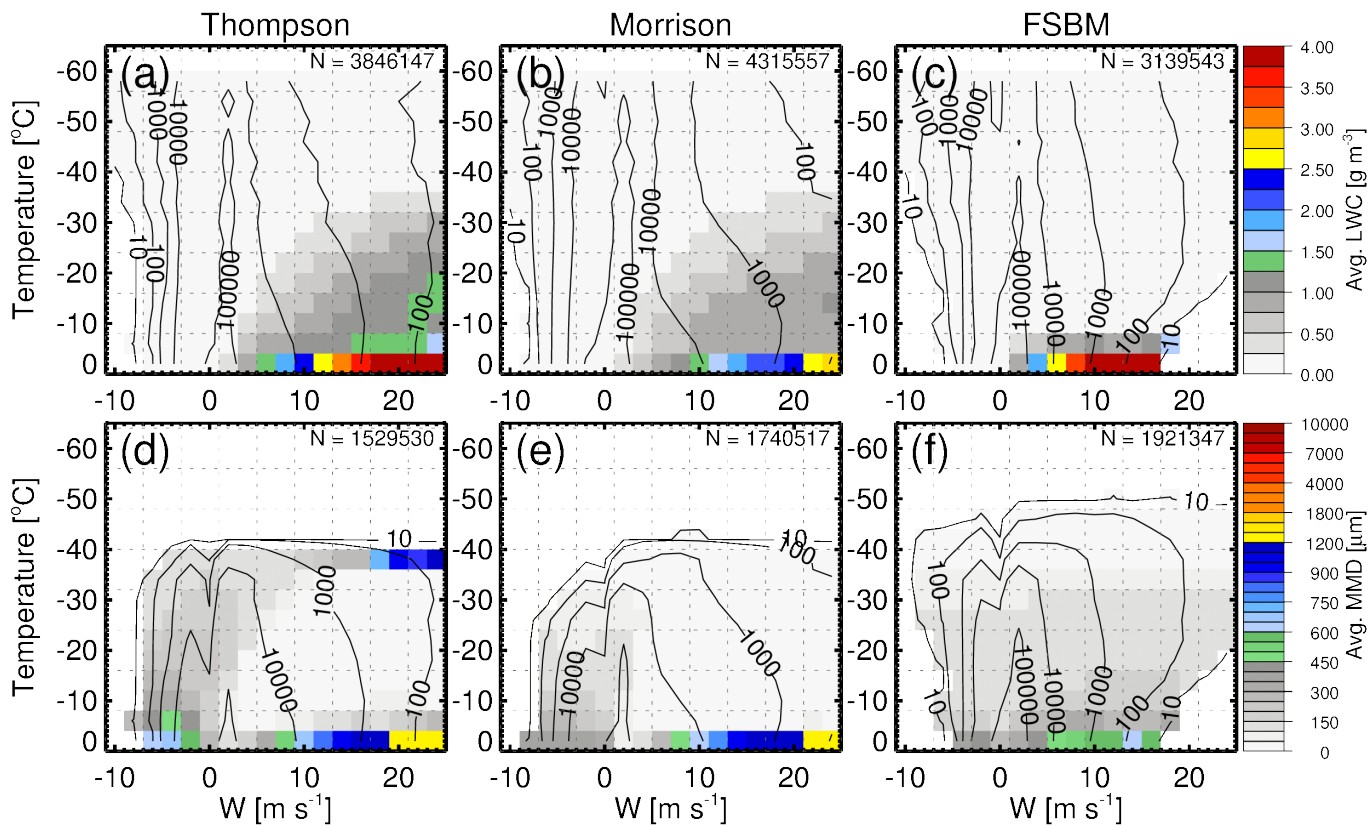

**Figure 14.** As in Fig. 11, but for liquid water content (LWC) color-filled in (a)-(c) and average liquid MMDs color-filled in (d)-(f).

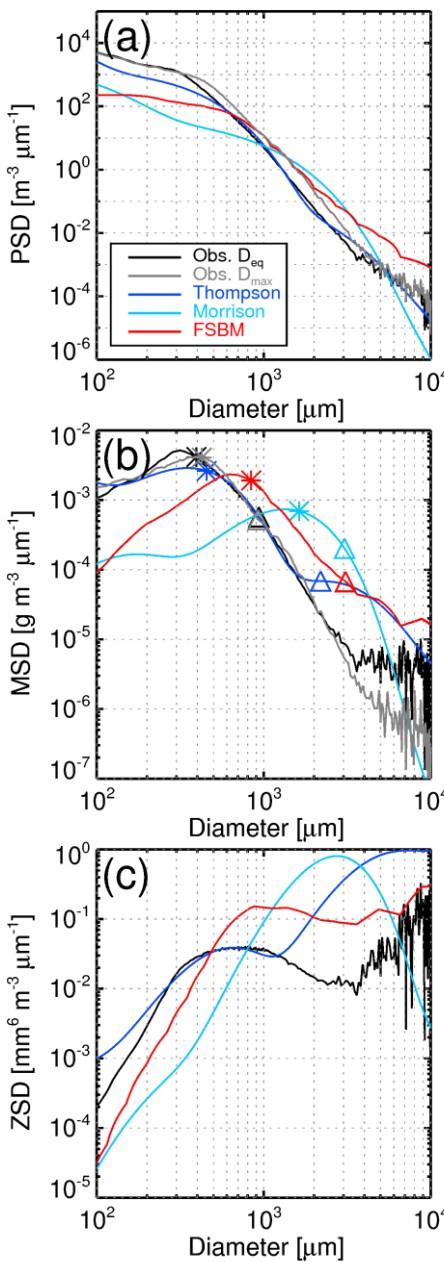

**Figure 15.** Observed and simulated composite average (a) PSDs, (b) MSDs, and (c) ZSDs for TWC between 2 and 2.5 g m$^{-3}$ and temperatures between -32 °C and -40 °C, where TWC is the integrated mass in diameters less than the observational maximum of 12.85 mm. Observed distributions for the D$_{eq}$ diameter definition are in black and for the D$_{max}$ definition are in grey. Thompson is in dark blue, Morrison in cyan, and FSBM in red. Asterisks and triangles overplotted on the MSDs in (b) are the MMD and 90% MD, respectively. Note that both axes are logarithmic.