# Peer review of "A ubiquitous ice size bias in simulations of tropical deep convection"

_Atmospheric Chemistry and Physics, 2017_

## Referee Comment (RC1) · Anonymous Referee #1 · 4 Apr 2017

This manuscript compares ice-phase particle sizes simulated by the WRF model with aircraft measurements during the recent HAIC-HIWC field campaign. The three different microphysical schemes used in WRF all show larger mass median diameters when compared with the observations. Many previous studies have demonstrated the same bias. However, uncertainties exist as to what extent the bias is caused by model errors in dynamics (e.g., overestimation of the updraft velocity), bulk microphysical schemes' assumptions in particle size distributions, and the uncertainties or misrepresentation of microphysical processes. The unique aspect of the HAIC-HIWC field campaign is that it simultaneously measures particle sizes and vertical air velocity. The manuscript compares the observations and model simulations by stratifying MMD and TWC with w, in an effort to separate errors in dynamics from microphysics. Uncertainties in bulk microphysical schemes' PSD assumptions are also investigated by comparing two dif-

ferent bulk microphysical schemes and an explicit bin microphysical scheme, which does not have any PSD assumption. This study is scientifically sound and utilizes data from a new field campaign. However, the presentation quality is very poor. So much so that two of the figures (Fig. 4 and Fig. 9) are missing from the manuscript. Fig. 3 is duplicated in Fig. 4, and Fig. 8 is duplicated in Fig. 9. This is unacceptable for any publication.

Specific comments:

In addition to the 2 missing figures, some figures will need to be reorganized/combined in order to facilitate comparisons and discussions. Some of the descriptions are difficult to understand, as listed below. I hope the authors will consider rewrite them to improve readability.

1. The manuscript compared observed TWC and MMD stratified by w and T/height. Ostensibly missing is the comparisons of the total number concentration. The total number concentration should be the most accurate observation and is predicted in all 3 microphysical schemes. Comparison of Ntotal stratified with w and T and/or TWC will provide more insights in model errors. In a generalized microphysical framework, one-moment schemes solve mass equation (first order of mass); two-moment schemes in general solve both mass and number concentration (zeroth and first order of mass). There are also three-moment schemes which solve the zeroth, first and another higher order variable, usually the second moment of mass. Comparisons of the simulated total numbers with observations is essential for all microphysical scheme validations. It can also be carried out in high confidence especially when using in-situ data. I suggest that the authors add $N\_total$ with respect to both w and TWC in Fig. 6. And add corresponding $N\_total$ in model simulation plots.

2. Fig. 7, 8 and 9 can be combined to a 3x3 panel figure for easier comparison and discussion.

3. The color scheme in Fig. 7-9 is also confusing. The convention is that blue has

smaller value than red. I was initially confused by the fact that red represents negative and blue positive. I'd suggest the authors to flip the color scheme in these figures.

4. What is the rationale of using only data with w>1m/s? Can data with w<1m/s be used for model comparisons, too? Will it lead to the same conclusion?

5. The biggest problem with the observation is its sampling bias. Due to safety concerns, the airplane must avoid lightning and area with radar reflectivity above 40 dBZ. The authors mentioned this error, but didn't do anything to quantify it. Another problem is that all data from different events are used indiscriminately to compare with a single case simulation. The author should add some analyses to address these uncertainties. For example, Fig. 1 seems to show that on Feb. 18, the airplane sampling might be on the weaker part of the system. This case also seems to have the warmest Tb among the four cases shown. Is this true? Does samplings with high w mainly come from other cases? One way of determine the bias is by plotting sampling sizes for Feb. 18 case only, and compare the result with the same plot using all samples, on a T-w and/or T-TWC plot. Another possibility to address the sampling bias is to combine the 3D C-Pol data with Feb. 18 sampling. For example, one could plot C-POL sampling sizes in T-dBZ space. Then use observed PSD to calculate dBZ for in-situ measurements, and plot sample sizes on a T-dBZ diagram, too. This could roughly show how much/what type of biases existed in airplane sampling.

6. The comparisons are made for model output between 18Z on the 18th to 00Z on the 19th, because this time period was considered to represents mature and dissipating stage of the MCS, according to the manuscript. However, Fig. 5 compares reflectivity at 16Z, which is outside the window for PSD comparisons. I suggest the authors to plot C-Pol radar reflectivity CFADs (Yuter, 1995, Mon. Wea. Rev., Vol123, P1941-1963) for the same 6-hour period. Then compare it with CFADs of model simulations.

Technical Corrections:

1. Please put correct Fig. 4 and Fig. 9.

2. P1, L24: "….., differences with observations for a given particle size vary greatly between schemes." I don't understand this statement. Please rephrase.

3. P4, L12: "Uncertainty in w calculations is estimated at ~1ms-1". Please give reference.

4. P5, first paragraph: Can the authors list the values of alpha and beta derived from the observations. They can be used to compare with model parameters, and are essential for reproducing the results (e.g., deriving MMD) shown in this study.

5. P5, L5: Deq is defined here as "area equivalent diameter", but later in eqn. 5 is used as "melted equivalent diameter". This is confusing.

6. P5, L27: ACCESS-R is used as the initial and boundary condition, not as "large-scale forcing", according to the conventional term usage.

7. P7 L2, where is the citation for CCN concentration? How high is the boundary layer?

8. P7, L9, "(not shown)" Can the authors show it, perhaps in the supplemental material. This is import if we want to use one case to represent all simulations.

9. P7, L15: The last sentence needs to be broken into two. The sentence has two unrelated issues about inner domain and total simulation time, if I understood it correctly.

10. P10, last paragraph: The discussion of Fig. 10 needs clarification. I guess the first question I have is: why use 90%? Would 50% do? Why or why not? Also, the descriptions are hard to comprehend. Please make an effort to clarify it.

11. P13, L30: "The majority of graupel at T>8C is formed by freezing raindrops". Can you give references and/or supporting evidences? My understanding is that this is only true when updraft velocity is high. Otherwise riming could be the dominant process.

12. P15, L1: "distribution tails" could mean tails at both small and large size end. May be change it to "large size tails"?

13. Fig. 11 to 16: Can the x-axis be extended beyond 25 m/s to include higher w simulated in the model? A line can be added to indicate the observation range of 25m/s. This will give a full picture and help the readers better understand model differences and their related processes discussed in the paper.

---

## Referee Comment (RC2) · Anonymous Referee #2 · 28 Apr 2017

This manuscript compares the hydrometeor sizes as functions of total water content and vertical velocity from simulations using two bulk and one bin schemes with observations from a mature to decaying tropical mesoscale convective system on Feb. 18 2014 during the HAIC-HIWC field campaign to investigate how and why models overestimate the radar reflectivity and underestimate the high total water content in higher portion of tropical convective storms. This study shows that all scheme overestimate the mean mass diameter of hydrometeors by producing too much mass at large particle diameters due to the assumed PSD function, the mass-size relationship, the species partitioning and the parameterized microphysical processes.

In general, the approach to compare model results and observations by constraining environmental condition is somewhat novel and effective and the findings are interesting that justifies its publication in ACP. However, the organization of the texts and the

figures should be improved and the following points should be addressed before it is accepted for publication.

Major points:

1. A table showing the density and mass-size relationships of all hydrometeor species should be provided for FSBM. I would also suggest putting terminal velocity and mass relationships to all three tables. As mentioned in the manuscript, sedimentation process also causes the different behaviors of the overestimated hydrometeor size. Some related studies (terminal velocity impact on cloud and precipitation structure) should be cited too.

2. The upper limit of the Deq of hydrometeors in FSBM is less than 10 mm due to its relatively fewer bin numbers (33 vs. 36 from many other bin schemes). How come the PSD, MSD and ZSD in Fig. 17 showing sizes reaching 10 mm for FSBM?

3. Is the C-POL reflectivity data gridded? It will be tricky to show plan view just at 2.5 km of the C-POL reflectivity in Fig. 3(a) if the data is not gridded. What is the data sample size of the observed reflectivity for each level shown on Fig. 5? The unevenness of the sample size above and away from the radar can cause bias in the observed profile. How was the reflectivity calculated for each simulation? What wavelength was assumed in the model reflectivity calculation? Were mass-size relationships associated with particular schemes or the water equivalent diameter used to calculate the model reflectivity? How are the partially melted particles coated with water are treated in the calculation? Different ways of calculating the reflectivity will provide very different results. Some discussions about the uncertainties of the model reflectivity should be provided.

4. Figure 4 is the same as Fig. 3 and Fig. 9 is the same as Fig. 8. It is hard to follow the discussions about these figures.

5. It will be good to plot PSD of liquid and ice particles separately in Figs. 11 and 12.

6. The last sentence in the abstract is too strong. It is hard to imagine current micro-physics schemes will uniformly produce a high bias in reflectivity.

Minor points:

1. It will be good to also mention the lower size limit of the OAPs in page 4 line 15.

2. 2. The threshold for simulated condensation mass mixing ratio is too small at 10-12 kg kg-1. Using this threshold may introduce grid points with unrealistic results. 10-6 kg kg-1 should be a good threshold for the analysis.

3. You probably referred to Fig. 6b in line 14 on page 12.

4. Page 1, line 16: using "different" microphysics. . .

---

## Author Comment (AC1) · 13 Jun 2017

We thank the reviewer for their thoughtful comments. We feel that they have improved the manuscript, and we hope that our revisions satisfy any concerns. Please note that any references to page or line numbers in our "changes to manuscript" responses refer to the final revised version of the manuscript and reference to any figure numbers in our "changes to manuscript" responses refer to revised figure numbers. Figures included for responses below are referenced as Fig. R1-R5. Note that although all 5 response figures (R1-R5) may not be referenced in this particular response, they are discussed in responses to reviews by both ACP referees. All 5 response figures are included in response to both anonymous referees for consistency of discussion. Figures added to supplemental material are referenced as Figures S1-S4.

[Figure]

Specific Comments:

1. Comment: The manuscript compared observed TWC and MMD stratified by w and T/height. Ostensibly missing is the comparisons of the total number concentration. The total number concentration should be the most accurate observation and is predicted in all 3 microphysical schemes. Comparison of N_total stratified with w and T and/or TWC will provide more insights in model errors. In a generalized microphysical framework, one moment schemes solve mass equation (first order of mass); two-moment schemes in general solve both mass and number concentration (zeroth and first order of mass). There are also three-moment schemes which solve the zeroth, first and another higher order variable, usually the second moment of mass. Comparisons of the simulated total numbers with observations is essential for all microphysical scheme validations. It can also be carried out in high confidence especially when using in-situ data. I suggest that the authors add N_total with respect to both w and TWC in Fig. 6. And add corresponding N_total in model simulation plots.

Response: We agree that total number concentrations are certainly important for microphysics scheme evaluation, however they were not analyzed in detail for two reasons. First, comparisons are really only useful when liquid and ice are separated because liquid dominates the total number concentration when present (e.g., a typical cloud droplet concentration is 100,000 L-1 whereas a high ice concentration is 100 L-1). The issue is that liquid and ice number concentrations are not separated in measurements. Additionally, liquid in the form of small cloud droplets are common in mixed phase updrafts of the bulk schemes (see Figure 14), and will dominate the number concentration in these regions. Apart from this simulated liquid likely often being erroneous (e.g., from not maintaining liquid supersaturations in the bulk schemes), the cloud droplet number concentration in the bulk schemes is held constant and there are no constraints on the CCN PSD in FSBM other than that this event occurs in a clean, tropical maritime air mass. Therefore, liquid needs to be removed from any number concentration comparisons, which is easy in simulations, but difficult in measurements.

[Figure]

Second, the measured total number concentrations have significant uncertainty. Figure R1, provided by coauthor Alfons Schwarzenboeck, shows differences in three 1-minute composite PSDs (resulting from 5-second samples) using derivation techniques from 3 different institutions. Consistent differences in number concentrations can reach an order of magnitude for sizes below ∼150 $\mu$m (note the logarithmic ordinate scaling). These small particles control the total number concentration, but quantifying the observational number concentration uncertainty is a nontrivial task, and doing so is beyond the scope of this study. The large differences in PSDs shown in Figure R1 partly result from different techniques used to filter out shattered particles. TWC and MMD measurement have greater certainty than number concentration since the IKP2 measures TWC plus water vapor, where the subtraction of water vapor is the primary source of uncertainty, while MMD is not strongly impacted by differences in particle numbers at diameters smaller than 150 $\mu$m. Essentially, mass as a higher moment of the PSD than number concentration is much less impacted by small particles that are the basis for uncertainty. For example, Figure 7 of Leroy et al. (2017) shows that the mean 15% MD as a function of TWC for any given flight is never smaller than 100 $\mu$m.

Changes to manuscript: We have added language on P5 L16-19 in the manuscript to state why total number concentrations are neglected.

2. Comment: Fig. 7, 8 and 9 can be combined to a 3x3 panel figure for easier comparison and discussion.

Response: We agree that combining these 3 figures into one figure facilitates easier discussion.

Changes to Manuscript: Former Figures 7, 8, and 9 have been combined into a 3x3 panel (current Figure 7)

3. Comment: The color scheme in Fig. 7-9 is also confusing. The convention is that blue has smaller value than red. I was initially confused by the fact that red represents negative and blue positive. I'd suggest the authors to flip the color scheme in these

figures.

Response: We agree that the convention is to use red for positive values and blue for negative values.

Changes to Manuscript: All relative difference plots now use red for positive values and blue for negative values (see current Figures 7, 8, S2, S3, and S4).

4. Comment: What is the rationale of using only data with w>1m/s? Can data with w< 1 m/s be used for model comparisons, too? Will it lead to the same conclusion?

Response: The original reasoning for using only updrafts was that high IWC regions targeted during the HAIC-HIWC field campaign are typically found in updrafts or regions connected to nearby convective cores, and are hypothesized to form within updrafts and regions detrained out of convective updrafts (see Lawson et al., 1998). However, we agree that this study would be more conclusive by including downdrafts and quiescent regions. This also provides a more statistically robust observational sample size for many significant TWC values (total sample size has increased approximately 8-fold, see Figure 6). An additional constraint was added (see response to Minor Point #2 by Reviewer 2) by which only simulated and observed points with TWC > 0.1 g m-3 are included. This value is similar to the expected uncertainty in measured TWC at -40°C (see discussion on P4 L28-33).

Changes to Manuscript: The dataset now includes all vertical velocity values with only the following constraints: (1) -60 °C < T < 0 °C and (2) TWC > 0.1 g m-3.

5. Comment: The biggest problem with the observation is its sampling bias. Due to safety concerns, the airplane must avoid lightning and area with radar reflectivity above 40 dBZ. The authors mentioned this error, but didn't do anything to quantify it. Another problem is that all data from different events are used indiscriminately to compare with a single case simulation. The author should add some analyses to address these uncertainties. For example, Fig. 1 seems to show that on Feb. 18, the airplane sampling

might be on the weaker part of the system. This case also seems to have the warmest Tb among the four cases shown. Is this true? Does samplings with high w mainly come from other cases? One way of determine the bias is by plotting sampling sizes for Feb. 18 case only, and compare the result with the same plot using all samples, on a T-w and/or T-TWC plot. Another possibility to address the sampling bias is to combine the 3D C-Pol data with Feb. 18 sampling. For example, one could plot C-POL sampling sizes in T-dBZ space. Then use observed PSD to calculate dBZ for in-situ measurements, and plot sample sizes on a T-dBZ diagram, too. This could roughly show how much/what type of biases existed in airplane sampling.

Response: There are several difficulties in quantifying the observational sampling bias. Although Fig. 1 shows that the aircraft sampled warmer brightness temperatures in the Feb. 18 system compared to others, this was necessary to observe convective cores within range of the C-POL radar, cores that still had cloud tops reaching up to 14-km altitude as observed by the onboard RASTA W-band radar. It is possible to compare C-POL reflectivity within the flight path to all C-POL reflectivities over the limited range of temperatures observed during this flight, but the most intense convection during this case was before the flight. During the flight, no cells were avoided within the region of interest (the C-POL domain). Therefore, the flight during this event is not representative of avoidance of high reflectivity regions or lightning, and trustworthy Rayleigh reflectivity was not observed for other cases. Additionally, using full 3-D C-POL reflectivity assumes that the aircraft would observe all regions equally, but this is not true, since the aircraft was specifically targeting regions thought to possibly have high IWC subjectively based on both pilot's radar data and satellite data. Therefore, there is no objective way to quantify this sampling bias in terms of reflectivity or any other variable. The impact of this bias is limited to the extent possible by controlling for TWC and vertical velocity, and is also the reason that the minimum 90% MD is examined, since it is not impacted by this sampling bias.

Additionally, Flight 23 on Feb. 18 is not representative of other flights. Attached is a

PDF (Figure R2) showing the contribution of each flight to the dataset used for this study. The 4 simulated events are indicated. On the right ordinate, Figure R2 shows the mean and standard deviation of temperature. Note that although a single flight may have sampled multiple temperature levels, each discrete flight leg flew at an approximately constant temperature level. This PDF signifies the need to use observations from the entire field campaign in order to properly stratify variables by temperature like the data shown in Figure 6. The Feb. 18 case (Flight 23) contains only ∼7-8% of the total samples, the majority of which remain between -10 and -20 °C. This PDF also show that using data from only the Feb. 18 event would not yield information on relatively colder temperatures, where most observations exist.

Figures S2-S4 were added in the supplemental material to address the concern of comparing one simulated event with observations from all field campaign events. These figures are similar to Figure 7, but show relative differences in observed and simulated TWC and MMD as a function of w-T and/or TWC-T bins for 3 additional events (Jan. 23, Feb. 2-3, and Feb. 7 events in Figures S2-S4, respectively) simulated with the bulk schemes. Note that FSBM was only run for the Feb. 18 simulation due to its high computational expense. These supplemental figures show that when controlling for temperature, TWC, and/or w, every simulated event yields very similar differences with observations in MMD-w-T-TWC space (where observations are taken from all flights), particularly in the case of MMDs. While there is certainly variability in the peak vertical velocities of simulated events, we control for vertical velocity, and thus this variability does not impact our results.

We note that the intention of focusing on the Feb. 18 event simulations is that radar (C-POL) observations were available for this event alone, and it contains most of the high TWC and high w observations at relatively warm temperatures. Figures S2-S4 clearly show that simulating a single event with different microphysics schemes yields much larger differences than simulating different events with the same microphysics scheme, at least in the phase space considered in this study.

Changes to Manuscript: Figure R2 has been added to the supplemental material as Figure S1 for extra justification of our comparison methodology. Figure S1 is discussed on P4 L7-8 and P7 L19-22. Figures S2-S4 have been added to the supplemental material to show that using a single simulated event is sufficient for comparison with all Darwin observations in the MMD-w-T-TWC phase space considered in this study. A short discussion of Figures S2-S4 is now provided on P11 L32 – P12 L3.

6. Comment: The comparisons are made for model output between 18Z on the 18th to 00Z on the 19th, because this time period was considered to represents mature and dissipating stage of the MCS, according to the manuscript. However, Fig. 5 compares reflectivity at 16Z, which is outside the window for PSD comparisons. I suggest the authors to plot C-Pol radar reflectivity CFADs (Yuter, 1995, Mon. Wea. Rev., Vol123, P1941-1963) for the same 6-hour period. Then compare it with CFADs of model simulations.

Response: Figure 5 has been revised such that it now shows only the 99th percentile to account for strictly convective radar reflectivities since there is a significant amount of stratiform precipitation across the domain (see Figures 3 and 4). Figure 5a shows the 99th percentile for a time period between 12Z and 18Z on the 18th and Figure 5c shows the 99th percentile profile between 18Z on the 18th and 00Z on the 19th. Figures 5b and 5d show sample sizes normalized by domain area (C-POL domain is smaller than the model domain) for the 12Z-18Z and 18Z-00Z time periods, respectively. These two time periods are shown because the most intense convection observed by C-POL was between 12Z-18Z, but the flight took place between 18Z-00Z. Although the most intense convection was not in the C-POL domain during the 18Z-00Z analysis period, it is clear that all simulations produce significantly higher reflectivities aloft compared to C-POL, no matter the time period selected for comparison. 99th percentile profiles and domain-normalized sample sizes are still used as opposed to CFADs. Using percentile profiles on one figure allows quantification of differences between observations and simulations that is not possible in CFADs. Moreover, the high IWC regions targeted by

the campaign aircraft are generally found in and around convective cores, and regions of modest stratiform reflectivity that would dominate CFADs were not of interest. The large simulated reflectivity biases aloft that are being explored in this study are commonly associated with convective regions. This is the bias being investigated with the in situ microphysical and kinematic datasets, and thus comparison of high percentile profiles (e.g. 99th) is more appropriate than CFADs.

Changes to Manuscript: Importantly, we note that we have used an updated C-POL dataset provided by Alain Protat who has been added as a co-author. We have also gridded these C-POL radial sweep files ourselves to match the simulation horizontal grid spacing (1 km) and found a ∼2.5 dB increase, which is why the cross-section figures (Figures 3 and 4) change from the original version, however this increase is not central to any of our conclusions. Figure 5 has been revised as discussed in the response above. Discussion of Figure 5 has been revised on P10 L4-15.

Technical Corrections:

1. Comment: Please put correct Fig. 4 and Fig. 9.

Changes to manuscript: The correct version of Figure 4 is now in place. The correct version of former Figure 9 is now represented in the bottom panel of Figure 7 (g-i), as recommended in Specific Comment #2.

2. Comment: P1, L24: ". . .., differences with observations for a given particle size vary greatly between schemes." I don't understand this statement. Please rephrase.

Changes to manuscript: Clarified this statement on P1 L22-24.

3. Comment: P4, L12: "Uncertainty in w calculations is estimated at ∼1 ms-1". Please give reference.

Changes to manuscript: We added a reference to Jorgensen and LeMone (1989) on P4 L13.

4. Comment: P5, first paragraph: Can the authors list the values of alpha and beta derived from the observations. They can be used to compare with model parameters, and are essential for reproducing the results (e.g., deriving MMD) shown in this study.

Response: Observed values of $\alpha$ and $\beta$ are not constant in the observational dataset, as explained on P5 L4-6 and in much more detail in Leroy et al. (2016). For this reason, the > 16,000 m-D coefficient pairs used in the current study are not listed. The $\beta$ parameter changes during flight since it is related to the exponents of area-size and perimeter-size power laws that are derived directly from OAP images for every 5-second flight sample. The $\alpha$ parameter varies during flight as it is constrained by independent TWC retrievals. Additionally, simulated $\alpha$ and $\beta$ parameters that are comparable to observations are not possible since observations include all hydrometeor types in derivation of the power law, while simulations have multiple hydrometeor species, each with different $\alpha$ and $\beta$ parameters, and combining these species to produce the mass size distribution means that the simulated mass-size relationship no longer follows a power law.

Changes to manuscript: Due to possible confusion, the sentence on P16 L20-23 has been modified to indicate that each 5-second flight sample uses a single m-D power law (not the entire dataset using the same power law/coefficients), and that the assumption that the distribution of mass with diameter is well represented by a power law may not be a good assumption in some situations. However, analyzing composite distributions may alleviate this issue to some degree.

5. Comment: P5, L5: Deq is defined here as "area equivalent diameter", but later in eqn. 5 is used as "melted equivalent diameter". This is confusing.

Changes to manuscript: All simulated Deq values are now referenced as Deq,melt in the computation of reflectivity where equivalent melted diameter is used, whereas Deq is now explicitly used to refer to the 2D area equivalent ice diameter, defined as the diameter of a circle with the same area as particle images from the OAPs.

6. Comment: P5, L27: ACCESS-R is used as the initial and boundary condition, not as "largescale forcing", according to the conventional term usage.

Changes to manuscript: Changed "large-scale forcing" to "initial and boundary conditions" on P6 L2-3.

7. Comment: P7 L2, where is the citation for CCN concentration? How high is the boundary layer?

Response: Perhaps there was some confusion in the values listed that give a vertical representation of the initial aerosol concentration profile in the FSBM scheme. These values are the default values used by FSBM to represent a maritime (pristine) air mass (set using FCCNR_MAR in module_mp_fast_sbm.F, the WRF microphysics module for the FSBM scheme). The text has been changed to make it clear that these values are not observed values, since there are none near Darwin for this field campaign, but rather the default values used by FSBM for a maritime air mass.

Changes to manuscript: We have changed "in the boundary layer" to "near the surface" on P7 L8-10 to note that this initial condition of CCN does not depend on boundary layer height.

8. Comment: P7, L9, "(not shown)" Can the authors show it, perhaps in the supplemental material. This is import if we want to use one case to represent all simulations.

Changes to manuscript: Added Figures S2-S4 in supplemental material showing differences in TWC and MMD as a function of w, TWC, and T for the other simulated events using the Morrison and Thompson scheme. See response to Major Point #5 for more information.

9. Comment: P7, L15: The last sentence needs to be broken into two. The sentence has two unrelated issues about inner domain and total simulation time, if I understood it correctly.

Changes to manuscript: The referenced sentence was broken into 2 and moved to the

middle of the paragraph on P7 L22-24.

10. Comment: P10, last paragraph: The discussion of Fig. 10 needs clarification. I guess the first question I have is: why use 90%? Would 50% do? Why or why not? Also, the descriptions are hard to comprehend. Please make an effort to clarify it.

Response: We have attempted to clarify the meaning and utility of the minimum 90% MD in the current study. The mass placed in the upper percentiles of the mass-size distribution (e.g. 90%) is typically from larger particles that may have been less frequently sampled by the aircraft due to the aforementioned sampling bias. However, the minimum 90% MD observed should not be impacted by this bias since it will be associated with a lack of large, dense particles and a lack of relatively high reflectivity or lightning. Therefore, if the simulation is unable to produce a single 90% MD that is as small as the observed minimum 90% MD for a given temperature, TWC, and vertical velocity, then we feel confident in declaring a model bias. This is particularly true because the simulations produce on average ~$10^2$ more samples for a given TWC-T or w-T bin compared to observations. So, despite 2 orders of magnitude larger sample sizes in a simulation, the simulation is unable to produce a single 90% MD as small as the observed minimum 90% MD in many TWC-T and w-T bins.

Changes to manuscript: Clarification of the 90% MD discussion has been made on P12 L10-19.

11. Comment: P13, L30: "The majority of graupel at T>8C is formed by freezing raindrops". Can you give references and/or supporting evidences? My understanding is that this is only true when updraft velocity is high. Otherwise riming could be the dominant process.

Response: You are correct that the dominant process is riming. We should have stated more clearly that the primary contributor to graupel production was the heterogenous freezing of raindrops due to the collision of rain and cloud ice, which is a riming process that involves freezing of lofted raindrops. This was analyzed in the Thompson scheme

and not in the FSBM scheme, however the raindrop sized MMDs and large LWCs between 0 and -4 °C in updrafts in Figure 14 indicate that the majority of the LWC is constituted by raindrops. In FSBM, by -8 °C, nearly all of the liquid is gone and most of the IWC is constituted by graupel (compare Figure 13 to Figure 12), which only decreases with decreasing temperature. This suggests that most of the graupel is being formed by raindrops that are freezing, likely heterogeneously through interactions with ice, as in the Thompson scheme. Since raindrops are smaller for T > -8 °C in FSBM, it intuitively makes sense that this contributes to smaller graupel sizes.

Changes to manuscript: Language has been changed on P15 L21-28 to clarify this and suggest this as a possible mechanism responsible for smaller graupel sizes rather than definitively concluding that it is the reason.

12. Comment: P15, L1: "distribution tails" could mean tails at both small and large size end. May be change it to "large size tails"?

Changes to manuscript: Added "at larger diameters" after "distribution tails" to clarify on P16 L25.

13. Comment: Fig. 11 to 16: Can the x-axis be extended beyond 25 m/s to include higher w simulated in the model? A line can be added to indicate the observation range of 25m/s. This will give a full picture and help the readers better understand model differences and their related processes discussed in the paper.

Response: The primary purpose of this study is to compare simulations with observations in order to establish potential model biases. Because observed vertical velocities do not exceed 25 m s-1, extending the x-axis would only allow comparison between different microphysics schemes, which is not a first-order objective in this study. The evaluation of differences between schemes in Figures 11-14 is primarily to provide possible explanations for biases with respect to observations. A more in-depth analysis of differences between simulations would be interesting, but we feel that it would extend the paper beyond a reasonable length for a single study.

**REFERENCES**

Lawson, R. P., Angus, L. J., and Heymsfield, A. J.: Cloud Particle Measurements in Thunderstorm Anvils and Possible Weather Threat to Aviation, J. Aircr., 35, 113-121, doi: 10.2514/2.2268, 1998.

Leroy, D., Fontaine, E., Schwarzenboeck, A., and Strapp, J. W.: Ice crystal sizes in high ice water content clouds. Part I: On the computation of median mass diameter from in situ measurements, J. Atmos. Oceanic Technol., 33, 2461–2476, doi: 10.1175/JTECH-D-15-0151.1, 2016.

Leroy, D., Fontaine, E., Schwarzenboeck, A., Strapp, J. W., Korolev, A., McFarquhar, G., Dupuy, R., Gourbeyre, C., Lilie, L., Protat, A., Delanoë, J., Dezitter, F., and Grandin, A.: Ice crystal sizes in high ice water content clouds. Part 2: Statistics of mass diameter percentiles in tropical convection observed during the HAIC/HIWC project, J. Atmos. Oceanic Technol., 34, 117-136, doi: 10.1175/JTECH-D-15-0246.1, 2017.

[Figure]

**Fig. 1.** PSDs from 3 different 1-minute composites (resulting from 5-second samples) showing PSDs derived from 3 different institutions. Number concentration is on the ordinate and diameter is on the abscissa.

[Figure]

**Fig. 2.** Distribution of flight samples for the Darwin HAIC-HIWC campaign by flight number (blue bars). Red diamonds show the mean temperature for each flight, with lines that show +/- one standard deviation.

[Figure]

**Fig. 3.** FSBM (a) mass , (b) density, and (c) terminal velocity as functions of diameter for liquid (solid), snow (dotted), and graupel (dashed).

[Figure]

**Fig. 4.** (a)-(c) Combined hydrometeor 10% MD, MMD, and 90% MD, respectively, as a function of w for T between -32 °C and -40 °C. Snow MDs in (d)-(f), graupel MDs in (g)-(i), and liquid MDs in (j)-(l).

**Fig. 5.** As in Figure R4, but for a temperature range between -8 °C and -16 °C.

[Figure]

---

## Author Comment (AC2) · 13 Jun 2017

We thank the reviewer for their insightful comments. We feel they have helped make our study more robust and hope that any concerns have been alleviated in our provided response. Please note that any references to page or line numbers in our "changes to manuscript" responses refer to the final revised version of the manuscript and reference to any figure numbers in our "changes to manuscript" responses refer to revised figure numbers. Figures included for responses below are referenced as Fig. R1-R5. Note that although all 5 response figures (R1-R5) may not be referenced in this particular response, they are discussed in responses to reviews by both ACP referees. All 5 response figures are included in response to both anonymous referees for consistency of discussion. Figures added to supplemental material are referenced as Figures S1-

[Figure]

S4.

Major Points

1. Comment: A table showing the density and mass-size relationships of all hydrometeor species should be provided for FSBM. I would also suggest putting terminal velocity and mass relationships to all three tables. As mentioned in the manuscript, sedimentation process also causes the different behaviors of the overestimated hydrometeor size. Some related studies (terminal velocity impact on cloud and precipitation structure) should be cited too.

Response: Table 3 has been added to list the mass-size relationship parameters and terminal velocity-size relationship for FSBM. Terminal velocity-size relationship parameters have also been added for Thompson and Morrison in Tables 1 and 2. Note that the terminal velocity-size relationships given for FSBM in Table 3 are listed as ranges. This is because the v-D relationships used in the study come directly from ASCII files located in the /WRFV3/run/ directory of V3.6.1 of WRF. Figure R3 shows mass, density, and terminal velocity as a function of diameter in panels (a)-(c), respectively, using the ASCII file data for each of the 33 size bins of each hydrometeor species. Updated analytical formulas are not provided in the literature for the relationships shown in Figure R3, and therefore, v-D relationship as well as the rho-D relationship for snow are listed as ranges rather than a formula.

Changes to manuscript: In addition to the inclusion of Table 3 and the inclusion of v-D relationship parameters for the Morrison and Thompson schemes, we have added language on P16 L4-8 to indicate that v-D relationships may impact model size biases presented in this study, but that analyzing this possible contribution is beyond the scope of the study. Relevant references have also been cited that examine impacts of sedimentation on cloud and precipitation structure.

2. Comment: The upper limit of the Deq of hydrometeors in FSBM is less than 10 mm due to its relatively fewer bin numbers (33 vs. 36 from many other bin schemes). How

come the PSD, MSD and ZSD in Fig. 17 showing sizes reaching 10 mm for FSBM?

Response: The upper diameter limit for snow in the FSBM scheme within WRF is actually 19.877 mm (9.9387 mm radius). This can be found by looking in the WRFV3/run/ directory for V3.6.1 in the ASCII file named "bulkradii.asc_s_0_03_0_9", which lists the corresponding bulk radius bin for each FSBM species mass bin. Although these snow particles are extremely large, aggregates in reality achieve these sizes, and the density is assumed to be low (35 kg m-3).

3. Comment: Is the C-POL reflectivity data gridded? It will be tricky to show plan view just at 2.5 km of the C-POL reflectivity in Fig. 3(a) if the data is not gridded. What is the data sample size of the observed reflectivity for each level shown on Fig. 5? The unevenness of the sample size above and away from the radar can cause bias in the observed profile. How was the reflectivity calculated for each simulation? What wavelength was assumed in the model reflectivity calculation? Were mass-size relationships associated with particular schemes or the water equivalent diameter used to calculate the model reflectivity? How are the partially melted particles coated with water are treated in the calculation? Different ways of calculating the reflectivity will provide very different results. Some discussions about the uncertainties of the model reflectivity should be provided.

Response: Yes, the reflectivity data from C-POL is Cartesian gridded. Figure 5 has been revised according to Specific Comment #6 by Reviewer 1. We have provided sample sizes, normalized by domain area, of the data used in Figure 5 for each time period shown. While the C-POL domain is smaller than the WRF domains, the number of grid points $\geq$ 5 dBZ per kmˆ2 is comparable between C-POL and WRF for the 12Z-18Z time period, while the C-POL sample size per kmˆ2 is larger than WRF for the 18Z-00Z time period. Additionally, the larger sample sizes used by analyzing 6-hr time periods (rather than a single time step that was previously shown) suggests that sample size limitations would not erase the extremely large difference in reflectivity aloft shown in the 99th percentile profiles of Figure 5.

Reflectivity calculations assume Rayleigh scattering, which is valid for comparison with the (5.5-cm) C-POL wavelength in this environment. This calculation follows Smith (1984), is commonly used in studies, and is consistent with PSDs and single particle properties assumed in each of the schemes. Rayleigh reflectivity is now output by default for several microphysics schemes in WRF including the Thompson and Morrison schemes used here. As discussed in Smith (1984), the estimated Rayleigh reflectivity for ice could be slightly off, but any potential error is much smaller than the very large differences between observations and simulations shown in Figure 5. The equation does indeed account for the m-D relationship via the melted equivalent diameter calculation (Equation 5), and thus the reflectivity is diagnosed to change with changes in the m-D relationship.

Partially melted particles coated with water are not explicitly represented in the microphysics schemes. The peak in reflectivity near the melting level in the bulk schemes shown in Figure 5 is the result of a very simple parameterization in the WRF computed reflectivity, which is implemented to represent the "bright band" caused by water coated melting ice particles. This peak does not occur for FSBM since FSBM reflectivity is computed offline by summing reflectivity contributions from each hydrometeor size bin without implementation of the melting layer algorithm. Although the reflectivity peak caused by water coated melting ice could have significant errors, this does not impact our conclusions drawn for regions above this melting layer, which is the region we focus on.

Changes to manuscript: Importantly, we note that we have used an updated C-POL dataset provided by Alain Protat who has been added as a co-author. We have also gridded these C-POL radial sweep files ourselves to match the simulation horizontal grid spacing (1 km) and found a ∼2.5 dB increase, which is why the cross-section figures (Figures 3 and 4) change from the original version, however this increase is not central to any of our conclusions. We have included statements that C-POL data is Cartesian gridded and uses 1-km horizontal and 500-m vertical grid spacing in the

manuscript on P9 L16-18. Figure 5 has been revised, as discussed in the response above and in Specific Comment #6 by Reviewer 1, and relevant discussion of Figure 5 has been revised on P10 L4-15 We have moved Equations 5-7 from Section 5.5 to the beginning of Section 5.1 (P9) to introduce the calculation of reflectivity when the subject is initially discussed.

We state that simulated reflectivity is calculated assuming Rayleigh scattering on P9 L14-15 and provide a brief explanation of why this is a good assumption.

Language has been added on P9 L9-13 to note the differences in reflectivity calculations between FSBM and bulk schemes and to note that the bulk schemes use a parameterization to treat reflectivity calculations of partially melted ice coated with water but FSBM does not.

4. Comment: Figure 4 is the same as Fig. 3 and Fig. 9 is the same as Fig. 8. It is hard to follow the discussions about these figures.

Changes to manuscript: The correct version of Figure 4 is now in place. The correct version of former Figure 9 is now represented in the bottom panel of Figure 7 (g-i), by which the old Figures 7-9 are now combined into a 3x3 panel that is the current Figure 7, as recommended in Specific Comment #2 by Reviewer 1.

5. Comment: It will be good to plot PSD of liquid and ice particles separately in Figs. 11 and 12.

Response: While separating these figures into liquid and ice is insightful from a model only perspective, we use the combined hydrometeor distributions because observations are not separated into liquid and ice species. Simulated liquid and ice MDs as a function of vertical velocity are shown in Figure R4 for T between -32 °C and -40 °C and Figure R5 for T between -8 °C and -16 °C, however they are difficult to interpret without the contribution of each species to the total mass for a given vertical velocity (e.g., liquid MDs can be very small, but not contribute much to the overall MDs if liquid

doesn't contribute much to the overall mass). This is the reason that we feel that it is better to draw inferences from Figures 11-14 in explaining some differences in (current) Figures 7-10. For example, Figure 14 shows that for temperatures between -8 °C and -16 °C, LWC can be significant in the bulk schemes for w > 5 m s-1 with corresponding very small MMDs, whereas LWC is insignificant in FSBM and MMDs are significantly larger. Snow and graupel MMDs in Figures 12-13 are large in all schemes for these temperatures, and therefore, it is known that cloud droplets must be causing the sharp decrease in 10% MD with increasing vertical velocity in bulk schemes in Figure 10.

6. Comment: The last sentence in the abstract is too strong. It is hard to imagine current microphysics schemes will uniformly produce a high bias in reflectivity.

Changes to Manuscript: Changed language on P1 L26-28 to indicate that this statement refers only to the microphysics schemes evaluated in the current study and not all microphysics schemes in general.

Minor Points

1. Comment: It will be good to also mention the lower size limit of the OAPs in page 4 line 15.

Changes to manuscript: Added lower limits of OAPs to P4 L16-17.

2. Comment: The threshold for simulated condensation mass mixing ratio is too small at 10-12 kg kg-1. Using this threshold may introduce grid points with unrealistic results. 10-6 kg kg-1 should be a good threshold for the analysis.

Changes to manuscript: All simulated and observed data now use a constraint of > 0.1 g m-3, which is the approximate TWC observational uncertainty at -40°C (see discussion in manuscript on P4 L29-33.

3. Comment: You probably referred to Fig. 6b in line 14 on page 12.

Changes to manuscript: Changed reference from Fig. 6c to Fig. 6b on P14 L6.

4. Comment: Page 1, line 16: using "different" microphysics. . .

Changes to manuscript: Changed "differing" to "different" on P1 L16.

[Figure]

[Figure]

**Fig. 1.** PSDs from 3 different 1-minute composites (resulting from 5-second samples) showing PSDs derived from 3 different institutions. Number concentration is on the ordinate and diameter is on the abscissa.

[Figure]

**Fig. 2.** Distribution of flight samples for the Darwin HAIC-HIWC campaign by flight number (blue bars). Red diamonds show the mean temperature for each flight, with lines that show +/- one standard deviation.

[Figure]

**Fig. 3.** FSBM (a) mass , (b) density, and (c) terminal velocity as functions of diameter for liquid
(solid), snow (dotted), and graupel (dashed).

[Figure]

**Fig. 4.** (a)-(c) Combined hydrometeor 10% MD, MMD, and 90% MD, respectively, as a function of w for T between -32 °C and -40 °C. Snow MDs in (d)-(f), graupel MDs in (g)-(i), and liquid MDs in (j)-(l).

**Fig. 5.** As in Figure R4, but for a temperature range between -8 °C and -16 °C